# Zonally Asymmetric Influences of the Quasi-Biennial Oscillation on Stratospheric Ozone

Wuke Wang[1, 2, 3, 4], Jin Hong[1], Ming Shangguan[5], Hongyue Wang[1], Wei Jiang[1], and Shuyun Zhao[1, 3, 4]

[1]Department of Atmospheric Science,China University of Geosciences,Wuhan,China
[2]Key Laboratory of Meteorological Disaster (KLME), Ministry of Education & Collaborative Innovation Center on Forecast and Evaluation of Meteorological Disasters (CIC-FEMD), Nanjing University of Information Science and Technology, Nanjing, China
[3]Research Centre for Complex Air Pollution of Hubei Province, Wuhan, China
[4]Centre for Severe Weather and Climate and Hydro-Geological Hazards, Wuhan, China
[5]School of Geography and Informaiton Engineering,China University of Geosciences,Wuhan,China

**Correspondence:** Ming Shangguan (shanggm@cug.edu.cn)

**Abstract.** The Quasi-Biennial Oscillation (QBO), as the dominant mode in the equatorial stratosphere, modulates the dynamical circulation as well as the distribution of trace gases in the stratosphere. While the zonal mean QBO signals in stratospheric ozone have been relatively well documented, the zonal (longitudinal) differences of the QBO ozone signals have been less studied. Using satellite based total column ozone (TCO) data from 1979 to 2020, zonal mean ozone data from 1984 to 2020,

three dimension ozone data from 2002 to 2020 as well as ERA5 reanalysis and model simulations from 1979 to 2020, we demonstrate that the influences of the QBO (using a QBO index at 20 hPa) on stratospheric ozone are zonally asymmetric. The global distribution of stratospheric ozone varies significantly during different QBO phases. During QBO westerly (QBOW) phases, the total ozone column (TCO) and stratospheric ozone are anomalously high in the tropics, while in the subtropics they are anomalously low over most of the areas, especially during the winter-spring of the respective hemisphere. This confirms

the results from previous studies. In the polar region, the TCO and stratospheric ozone (50-10 hPa) anomalies are seasonally dependent and zonally asymmetric: during boreal winter (DJF), positive anomalies of the TCO and stratospheric ozone are evident during QBOW over the regions from North America to the North Atlantic (120°W-30°E) while significant negative anomalies exist over other longitudes in the Arctic; in boreal autumn (SON), the TCO and stratospheric ozone are anomalously high from Greenland to Eurasia (60°W-120°E), but anomalously low in other regions over the Arctic; weak positive

TCO and stratospheric ozone anomalies exist over the South America sector (90°W-30°E) of the Antarctic while negative anomalies of the TCO and stratospheric ozone are seen in other longitudes. The consistent features of TCO and stratospheric ozone anomalies indicate that the QBO signals in TCO are mainly determined by the stratospheric ozone variations. Analysis of meteorological conditions indicates that the QBO ozone perturbations are mainly caused by dynamical transport and also influenced by chemical reactions associated with the corresponding temperature changes. QBO affects the geopotential

height and the polar vortex and subsequently the transport of ozone-rich air from lower latitudes to the polar region, which therefore influences the ozone concentrations over the polar region. The geopotential height anomalies associated with QBO

(QBOW-QBOE) are zonally asymmetric with clear wave-1 features, which indicates that QBO influences the polar vortex and stratospheric ozone mainly by modifying the wave number 1 activities.

## 1 Introduction

Ozone is one of the most important trace gases in the stratosphere (Solomon, 1999; WMO, 2018). The ozone layer in the stratosphere protects life on the Earth by absorbing Ultraviolet (UV) Radiation (Solomon, 1999; WMO, 2018). Because of its strong radiative effects, ozone determines the thermal structure of the stratosphere (e.g. Son et al., 2008; Kodera et al., 2016; WMO, 2018). Changes in ozone influence the temperature in the stratosphere significantly and subsequently modify the stratospheric circulation due to the thermal-dynamical balance (Son et al., 2008; Xie et al., 2016; Solomon et al., 2016; Banerjee et al., 2020). On the other hand, changes in the stratospheric circulation, may cause a redistribution of ozone in the stratosphere because of the dynamical transport (Tweedy et al., 2017; Coy et al., 2016; Wang et al., 2019). Therefore, processes like the El Niño–Southern Oscillation (ENSO) and the Quasi-Biennial Oscillation (QBO), which have important influences on the interannual variations of the stratospheric circulations, affect the stratospheric ozone significantly (e.g. Lee et al., 2010; Xie et al., 2014; Lu et al., 2019; Xie et al., 2020).

The influences of QBO, which is the dominant mode of interannual variability in the equatorial stratosphere (Baldwin et al., 2001), on ozone have been investigated by lots of studies (e.g. Randel and Wu, 1996; Lu et al., 2019; Xie et al., 2020; Zhang et al., 2021). QBO appears as alternating easterly and westerly winds that propagate down from the top (∼50 km) to the bottom (∼16 km) of the stratosphere, with a period of about 28 months (Baldwin et al., 2001; Anstey and Shepherd, 2014; Coy et al., 2016). Such changes in zonal winds modify the vertical propagation of planetary waves and influence the strength of the polar vortex as well as the Brewer-Dobson circulation (BDC) according to the Holton-Tan mechanism (Holton and Tan, 1980, 1982; Watson and Gray, 2014; Zhang et al., 2019; Baldwin et al., 2019) or the QBO implicit meridional circulation mechanism (Garfinkel et al., 2012; Elsbury et al., 2021), and therefore play an important role in determining the dynamical circulation in the whole stratosphere (Naoe and Shibata, 2010; Garfinkel and Hartmann, 2011a, b; Anstey and Shepherd, 2014; Andrews et al., 2019; Zhang et al., 2020). Changes in stratospheric circulation during different QBO phases subsequently influence the redistribution of stratospheric trace gases (Tweedy et al., 2017; Xie et al., 2020). For example, many studies have reported QBO signals in methane, water vapour and ozone (Randel and Wu, 1996; Tian et al., 2006; Lu et al., 2019; Tao et al., 2019).

QBO signals in total column ozone (TCO) have been reported over 30 years ago using observations from surface Dobson stations (Hamilton, 1989) and satellites (Bowman, 1989; Tung and Yang, 1994). During QBO westerly (QBOW) phases, TCO is anomalously high in the tropics but low in the extratropics (Bowman, 1989; Hamilton, 1989; Tung and Yang, 1994). Besides such features, a seasonal synchronization, i.e., ozone anomalies are only significant during winter-spring of each respective sphere in the extratropics, is also indicated. The vertical structure of QBO signals in stratospheric ozone is then investigated using satellite data (e.g. Hasebe, 1994; Randel and Wu, 1996) and model simulations (e.g. Butchart et al., 2003; Tian et al., 2006; Hansen et al., 2013). A double-peaked vertical structure of ozone variations associated with QBO is clear in both the tropics and midlatitudes, with one peak in the lower stratosphere (20-30 km) and the other in the middle stratosphere (30-

40 km). The details of where the peaks lay depend on the level used to define the QBO (we use a QBO index at 20 hPa in this study, see details in the Data and Methods Section). With a longer record of ozone observations as well as more other data available, studies further investigated the combined influences of the QBO and other processes like the ENSO on the stratospheric ozone (Lee et al., 2010; Xie et al., 2014; Lu et al., 2019; Xie et al., 2020). The role of chemical processes in determining the ozone QBO is also studied (Zhang et al., 2021). So far, the impacts of the QBO on the TCO and stratospheric

ozone have been relatively well documented. However, all the studies mentioned above focused on the zonal mean features. The global distribution of the TCO and stratospheric ozone anomalies related to the QBO has not been investigated to our understanding.

This study investigates the global distribution (both zonal and meridional) of ozone anomalies related to QBO using satellite observations, reanalysis data and model simulations. Details of the data and methods used in this study are described in Section

2. Results, including the influences of the QBO on total column ozone (TCO), zonal mean ozone, ozone at different latitudes and longitudes as well as a possible mechanism are provided in Section 3. In Section 4, we summarize the conclusions and give a discussion.

## 2    Data and Methods

### 2.1    Ozone data from Satellites

Three types of ozone products from the Copernicus Climate Change Service (C3S), including the Total Column Ozone (TCO, with dimensions of longitude and latitude), merged zonal mean product (with dimensions of latitude and altitude) from limb sensors and merged three dimension product (3-D data with dimensions of longitude, latitude and altitude) from limb sensors, are used in this study. The TCO data is from MSR2 (Multi-Sensor Reanalysis, version 2), which spans the period 1979-2020 with a horizontal resolution of $0.5° \times 0.5°$. The zonal mean product is available from 1984 to 2020, which merged ozone

data from limb sensors of ACE (Atmospheric Chemistry Experiment onboard SCISAT), GOMOS (Global Ozone Monitoring by Occultation of Stars onboard ENVISAT), MIPAS (Michelson Interferometer for Passive Atmospheric Sounding onboard ENVISAT), OMPS (Ozone Mapping Profiler Suite onboard NPP ), OSIRIS (Optical Spectrograph and InfraRed Imager System onboard ODIN), SAGE-2 (Stratospheric Aerosol and Gas Experiment II onboard ERBS) and SCIAMACHY (Scanning Imaging Absorption Spectrometer for Atmospheric Cartography on board ENVISAT). It has a meridional resolution of

10 ° and covers 41 vertical levels from 10 km to 50 km. The 3-D ozone products merged ozone data from limb sensors of GOMOS, MIPAS, OSIRIS and SCIAMACHY. This data has a horizontal resolution of $20° \times 10°$ (longitude $\times$ latitude) and a vertical resolution of 1 km from 10 km to 50 km, and is only available for the period 2002-2020. The data are accessible from the website: https://cds.climate.copernicus.eu/cdsapp#!/dataset/satellite-ozone-v1?tab=form (accessed on 16 February 2022). According to the product quality assessment done by Copernicus Climate Change Service, biases at the or-

der of -0.13±0.11% were reported for the MSR2 TCO data, compared to the ground-based Dobson measurements (https: //datastore.copernicus-climate.eu/documents/satellite-ozone/C3S_312b_Lot2.3.2.1_202002_PUGS_O3_v1.21.pdf). More details of the data can be found at https://cds.climate.copernicus.eu/cdsapp#!/dataset/satellite-ozone?tab=overview.

## 2.2 ERA5 reanalysis data

The ERA5 reanalysis, which is the newest generation reanalysis product produced by the ECMWF Integrated Forecast System, is used in this study. The original model has 137 hybrid sigma model levels from the surface to the model top at 0.01 hPa. The horizontal resolution of the model is about 31 km. ERA5 assimilates a large number types of observations, including newly reprocessed data sets, recent instruments and cell-pressure corrected SSU, improved bias correction for radiosondes, etc., into global estimates using a 4D-Var data assimilation system. Especially, ERA5 assimilated numerous satellite ozone observations, including OMI (Ozone Monitoring Instrument), SCIAMACHY, MIPAS, GOME, GOME-2 and MLS (Microwave Limb Sounder) etc. (Hersbach et al., 2020), which helps the ERA5 ozone data to have good agreement with observations in the stratosphere (Shangguan et al., 2019). ERA5 covers a record of the global atmosphere from 1950 to the present. The ozone and other meteorological parameters in ERA5 for the period 1979-2020 are used in this study. More details about the ERA5 reanalysis data can be found at the website:https://confluence.ecmwf.int/display/CKB/ERA5%3A+data+documentation.

## 2.3 Model simulations with CESM-WACCM

To confirm the QBO impacts on ozone, a pair of simulations with NCAR's CESM model (version 1.0.2) is employed in this study. The model is integrated with its fully coupled mode, with interactive ocean, land, sea ice and atmosphere processes. To better represent the stratospheric ozone, the atmospheric component of WACCM (version 4), with detailed stratospheric dynamics and chemistry is used. The model covers a vertical range from the surface to about 140 km with 66 vertical levels (Marsh et al., 2013). The horizontal resolution of the model is 1.9 °× 2.5 °(latitude × longitude) for the atmosphere and approximately 1 degree for the ocean.

We first conducted a control run (Natural run), which includes all natural forcing like solar variability, interactive ocean, volcanic aerosols and a nudged QBO (Matthes et al., 2010). The solar and volcanic aerosol forcing is derived from observations for the period 1955-2004 and then based on future projection (2004-2099) following the SPARC (Stratospheric Processes and their Role in Climate) CCMVal (Chemistry-Climate Model Validation Activity) REF-B2 scenarios (SPARC-CCMVal, 2010). The QBO forcing time series is determined from the observed climatology of 1953-2004 via filtered spectral decomposition of that climatology. This gives a set of Fourier coefficients that can be expanded for any day and year in the past and the future. Anthropogenic forcing like greenhouse gases (GHGs) and ozone depleting substances (ODSs) are set to constant 1960s conditions to better illustrate the natural variability. In addition to the Natural run, a NOQBO run (without QBO nudging) is also employed to do a comparison. Both of the simulations are integrated from 1955 to 2099. While all other configurations of the two model simulations are the same, the difference between these two simulations specifies exactly the influences of the QBO nudging.

## 2.4 Methods

A composite analysis based on the time series of a QBO index is used to investigate the influences of the QBO on ozone and meteorological parameters. Because of the downward propagating of easterly and westerly regimes, the phase of the QBO is

120 different while selecting QBO indices at different levels (Anstey and Shepherd, 2014). Here we follow the method as indicated by previous studies (Wallace et al., 1993; Randel et al., 2009; Wang et al., 2015), which applied an Empirical Orthogonal Function (EOF) analysis on the equatorial zonal wind in the stratosphere (70-10 hPa). The observed equatorial zonal winds are provided by the Free University of Berlin http://www.geo.fu-berlin.de/en/met/ag/strat/produkte/qbo/index.html. A pair of orthogonal principal components (PCs) can be obtained from this EOF analysis (Fig. 1). The first principal component (PC1)

of the EOF mode is synchronized with the 20 hPa equatorial zonal wind with a correlation coefficient of 0.99, while the second PC (PC2) is synchronized with the 50 hPa equatorial zonal wind with a correlation coefficient of 0.93. PC1 is selected as the QBO index in this study to indicate the influences of QBO on stratospheric ozone. This is due to 2 reasons: 1) PC1 is close to the middle stratosphere ($\sim$10 hPa), where the ozone mixing ratios are highest; 2) The sample size of QBOW and QBOE (QBO easterly) is nearly equal to each other, while the QBOW size is usually much larger than the QBOE size using PC2

(Fig. 1). We then define the QBO westerly (PC1 > 0.5 standard deviation, QBOW) and easterly (PC1 < 0.5 standard deviation, QBOE) phases based on PC1 (Fig. 1a). The QBO associated signals/anomalies can be obtained by the differences of focusing parameters between QBOW and QBOE phases (QBOW-QBOE). We also check the sensitivity of the results to the standard to define the QBO phases (e.g. using the 0 instead of 0.5 standard deviation of the QBO index) and the results persist. Some of the results using PC2 as the QBO index are also discussed in Section 4.

As indicated by previous studies, the zonal mean ozone anomalies associated with QBO are mainly caused by the vertical transportation (Randel and Wu, 1996), which is dominated by the corresponding changes in the meridional overturning circulation in the stratosphere, i.e. the Brewer-Dobson Circulation (BDC) (Butchart, 2014; Coy et al., 2016). To explain the vertical structure of the QBO signals in ozone, the vertical component of the BDC in the meridional plane is calculated using the Transformed Euler Mean (TEM) equation (Andrews et al., 1987) as follows:

$$\overline{w^*} = \overline{w} + \frac{1}{a cos\phi}(\frac{cos\phi \overline{v'\Theta'}}{\overline{\Theta}_z})_\phi \qquad (1)$$

Where $v$ and $w$ are the horizontal and vertical winds, $\phi$ denotes the geopotential height, $\Theta$ is the potential temperature, overbar and prime denote the zonal mean and deviation from the zonal mean, respectively.

## 3 Results

### 3.1 Influences of the QBO on Total Column Ozone

We first revisit the QBO impacts on the TCO using the newest satellite-based multi-sensor reanalysis data (MSR2) with a data record over 40 years (1979-2020). From the MSR2 data, monthly anomalies of TCO near the equator (10°S-10°N) are anomalously high during QBOW phases compared with QBOE. At the same time, negative anomalies of TCO can be seen from the subtropics (15°) to mid-latitude ($\sim$50°) in both hemispheres (Fig. 2a). This is consistent with the results of previous studies (Bowman, 1989; Hamilton, 1989; Tung and Yang, 1994). In addition, significant TCO anomalies are also seen in the

polar regions of both hemispheres. Positive TCO anomalies are evident during QBOW over the regions from Greenland to

western Eurasia (60°W-100°E) while significant negative TCO anomalies exist over other regions in the Arctic and subarctic. Positive anomalies of TCO exist over South America and the South Atlantic sector (90°W-0°E), while negative TCO anomalies are significant over other areas of the subantarctic. This indicates zonally asymmetric features of TCO anomalies related to the QBO (QBOW-QBOE), which has not been reported by previous literature to the best of our knowledge. Such QBO features are very similar in ERA5 (Fig. 2b) to that seen in MSR2, indicating that ERA5 has a very good representation of the TCO variations. The Natural simulation by the CESM-WACCM model represents the QBO anomalies in TCO pretty well. The spatial pattern of the TCO anomalies associated with QBO from the CESM-WACCM simulation as seen in Fig. 2c shows very good consistency with that from the MSR2 data (Fig. 2a) in most of the areas, except that the TCO anomalies in the Antarctic are all positive. The magnitude of the negative anomalies in the subtropics from the CESM-WACCM model simulation is also slightly larger than that from the MSR2 and ERA5. Without the QBO nudging, the QBO signals disappear in the NOQBO run (Fig. 2d). Because the only difference between the two model simulations is the QBO nudging and because the difference in the two composites is similar between the simulation and the observation, this result indicates that the differences in the two composites of the observed TCO are mostly due to QBO.

As mentioned in previous studies, there is a seasonal synchronization in QBO related TCO signals (Hamilton, 1989; Tung and Yang, 1994). TCO anomalies associated with QBO (QBOW-QBOE) are only significant during the winter-spring of each respective sphere in extratropics. We therefore checked the QBO in TCO during 4 different seasons based on the QBO index in each season (an example of the QBO index in boreal winter is shown in Fig. S1). Fig. 3 gives the global distribution of the TCO anomalies (QBOW-QBOE) in different seasons resulting from a composite analysis using the MSR2 data. In all 4 seasons, positive TCO anomalies are seen near the Equator. This is consistent with previous studies, which indicated that the TCO signals in the tropics are not seasonally dependent (Bowman, 1989). Also consistent with earlier results, the negative TCO anomalies are more significant during the winter-spring of each respective sphere in the extratropics. However, some new features of the QBO signals (QBOW-QBOE) are found in our analysis of the global distribution: the QBO anomalies in TCO over the subarctic in boreal winter are zonally asymmetric, with positive anomalies over the regions of North America and North Atlantic (120°W-30°E) and negative anomalies over other regions; some TCO signals are also significant during boreal autumn in the northern hemisphere (NH) with zonally asymmetric features, i.e. positive TCO anomalies over North Atlantic and Eurasia (60°W-120°E) and negative TCO anomalies over the North American Arctic. To further illustrate the robustness of the results, the Natural and the NOQBO simulations are employed. The zonally asymmetric QBO signals (QBOW-QBOE) in the subarctic during DJF are also significant in the Natural simulation, but not evident in the NOQBO simulation (Fig. S2), which indicates a robust QBO impact. The zonally asymmetric features of the QBO signals in SON are less evident in the model, but more significant in MAM, which suggests that there are relatively large uncertainties in the two seasons.

### 3.2 Vertical Structure of the Influences of QBO on Stratospheric Ozone

The latitude-pressure cross-sections of ozone monthly anomalies associated with the QBO (QBOW-QBOE) from different data are shown in Fig. 4. Note that due to the data availability, results shown in Fig. 4 are based on data from 1985 to 2020.

From the merged satellite data, there are double-peaks of positive ozone anomalies over the equator during QBOW phases, which is consistent with previous studies (Randel and Wu, 1996; Xie et al., 2020; Zhang et al., 2021). In the extratropics (from subtropics to midlatitudes), there are also double-peak of the negative ozone anomalies, with one peak in the lower stratosphere (70-30 hPa) and the other in the middle stratosphere (20-5 hPa). QBO signals in ERA5 ozone (Fig. 4b) are in good agreement with the merged satellite data, except that the positive anomalies over the equator from ERA5 are not separated vertically.

The Natural run also shows good consistency with the satellite and ERA5 data (Fig. 4c), although the positive anomaly in the tropical upper stratosphere from the Natural run is located at a little higher altitude and extended higher than the observations, and the negative signals are extended higher up to the upper stratosphere in the extratropics. Without a QBO nudging, the signals are all blank indicating the robust contribution of the QBO nudging to the ozone signals in the stratosphere (Fig. 4d).

The vertical structure of QBO signals (QBOW-QBOE) in stratospheric ozone in different seasons based on the merged

satellite data (1985-2020) is shown in Fig. 5. Similar to the TCO signals, the equatorial ozone signals related to QBO are seasonally independent with positive ozone anomalies in the lower and middle stratosphere in all seasons. The QBO signals in ozone are mainly significant during the winter-spring of each respective sphere in the extratropics. Note that there are some positive ozone anomalies during QBOW phases over the Antarctic during JJA. To understand the reason for the vertical structure of ozone anomalies associated with the QBO, the corresponding changes in the vertical component of the BDC are

presented in Fig. 6. Seen from the climatological distribution, the ozone volume mixing ratio peaks in the tropics in the middle stratosphere at ∼10 hPa (Fig. S3). During QBOW phase, the vertical component of the BDC (w*) in the tropics is anomalously weak in the lower stratosphere and strong in the middle stratosphere in all seasons (Fig. 6), which transports ozone-poor air from the troposphere to the lower stratosphere and ozone-rich air from the middle stratosphere to upper levels and therefore leads to negative ozone anomalies in the lower stratosphere and positive ozone anomalies between 15 to 3 hPa (Fig. 5). The positive

anomalies in the tropical middle stratosphere (15-3 hPa) may also be related to the corresponding temperature changes since the gas-phase chemical reactions are temperature dependent (Solomon et al., 1985; Zhang et al., 2021). Cold temperature anomalies can be found in the tropical middle stratosphere (Fig. S4) because of the enhanced upwelling and subsequent dynamical cooling (Fig. 6), which may also contribute to the positive ozone anomalies in that region. Significant negative anomalies of w* can be seen in the subtropics and midlatitudes, especially during winter of each respective sphere (Fig. 6), which means enhanced

downward motion in these regions bringing ozone-poor air from upper levels to the middle stratosphere. Such negative ozone anomalies in the subtropical middle stratosphere may also be influenced by the warm temperature anomalies (Fig. S4).

### 3.3 Global distribution of QBO Signals in Stratospheric Ozone

We now discuss the global (both meridional and zonal) impact of the QBO on the stratospheric ozone, which has not been documented before to the best of our knowledge. Fig. 7 shows the global distribution of monthly ozone anomalies related to

215 the QBO (QBOW-QBOE) at ∼10 hPa by a composite analysis using different data sets (analyzed for the period 2002-2020 due to the data availability of the merged satellite data). From the merged satellite data, the spatial pattern of the ozone QBO is similar to that seen in the TCO anomalies as shown in Fig. 2. Zonally asymmetric features of the differences between QBOW and QBOE phases are seen in the polar regions of both hemispheres (Fig. 7a). The ERA5 data again show a consistency with

the satellite data except that the zonally asymmetric feature of the QBO signals in the Antarctic is not that obvious (Fig. 7b). The CESM-WACCM model, however, shows some differences with the satellite, with much weaker signals in the tropics. This is due to the shift of the ozone QBO to slightly higher altitudes as shown in Fig. 4c. Without a QBO nudging, the QBO signals seen in Fig. 7c disappear (Fig. 7d), indicating again a robust influence of the QBO on stratospheric ozone. Zonally asymmetric QBO signals in ozone can also be found in the lower stratosphere at 50 hPa (Fig. S5), with even more significance in the Arctic. This indicates that ozone changes in both the lower and the middle stratosphere contribute to the QBO signals in TCO, which is consistent with previous studies (Randel and Wu, 1996).

Fig. 8 gives the global distribution of QBO signals (QBOW-QBOE) in stratospheric ozone at ∼10 hPa in different seasons using the merged satellite data for the period 2002-2020. The zonally asymmetric features of the QBO signals in the polar regions during boreal winter in the Arctic and during boreal autumn (SON) in both hemispheres are more evident. Especially, clear and significant zonally asymmetric ozone anomalies associated with the QBO in both the Arctic and Antarctic regions can be seen in boreal autumn, which have not been documented in previous literature to the best of our knowledge. In the Arctic, positive ozone anomalies are seen in the North Atlantic and Eurasia (60°W-120°E) while negative anomalies can be found in other regions (Fig. 8c). In the Antarctic, positive anomalies are mainly located in the Atlantic sector while negative anomalies are more evident in the eastern hemisphere (Fig. 8c). It is also interesting that there are some positive ozone anomalies over the Antarctic during its winter, which was not mentioned by other studies. However, such positive ozone anomalies are very weak and not significant from the ERA5 data (Fig. S6), indicating that there are large uncertainties in these positive anomalies over the Antarctic. With a longer period of data (1979-2020), ozone anomalies over the Arctic during its winter are slightly more significant in the ERA5 data (Fig. S6d) compared to the satellite data (8d). Similar zonally asymmetric ozone anomalies associated with the QBO are also evident at 50 hPa (Fig. S7). Comparing the QBO signals in the stratosphere (at both 10 and 50 hPa) and the TCO anomalies shown in Fig. 3, the similar features indicate that the changes of ozone in the stratosphere play a dominant role in the TCO changes.

## 3.4 Dynamical Mechanism

As introduced in the Introduction, stratospheric ozone is related to complex photochemical processes which are temperature-dependent as well as dynamical transport. The meteorological conditions are therefore important for stratospheric ozone variations. To understand the possible mechanism of the QBO impacts on stratospheric ozone, meteorological parameters from the ERA5 reanalysis are analyzed for the period 1979-2020. Fig. 9 shows the global distribution of zonal wind (U) anomalies (shading) associated with the QBO (QBOW-QBOE) and the climatological U during QBOE (contour lines) in the middle stratosphere (at 10 hPa). While there are westerly anomalies in the tropics during QBOW, asymmetric wind anomalies are seen during winter of respective hemispheres in the extratropics. Easterly anomalies only exist over the Atlantic and the Mediterranean in the midlatitudes and over the North Pacific sector in the Arctic, while westerly anomalies are seen in other regions during boreal winter in the NH. In the SH, westerly anomalies can be seen over the western hemisphere of the Antarctic, while easterly anomalies exist over the eastern hemisphere during its winter-spring. The results are consistent with preceding studies (e.g. Anstey and Shepherd, 2014; Watson and Gray, 2014; Andrews et al., 2019), considering the different definition of QBO.

Temperature (T) anomalies at 10 hPa associated with the QBO (QBOW-QBOE) are given in Fig. 10. During QBOW phases, cold temperature anomalies are seen near the equator in all seasons. This is possibly related to the anomalously strong upwelling of the BDC in the tropics as seen in Fig. 6 and subsequent dynamical cooling. In the subtropics, warm temperature anomalies are dominant according to the enhanced downwelling of the BDC and subsequent dynamical warming. In the polar regions, cold anomalies are evident due to the weakening of the downwelling (Fig. 6) and subsequent less dynamical warming. Since the photochemical reactions related to ozone production are mostly temperature dependent, temperature anomalies may contribute to the ozone signals related to the QBO. Near the equator, cold temperature anomalies lead to slower ozone destruction and therefore contribute to positive ozone anomalies. For the same reason, warm temperature anomalies contribute to the negative ozone anomalies in the subtropics. In the polar region, the relationship between temperature and ozone is complicated, which is the same sign of changes in some regions (e.g., in the eastern hemisphere of the Arctic in DJF) but the opposite sign of changes in other regions (e.g., over the Antarctic in JJA). This is because that 10 hPa is a transition altitude from dynamical control at lower altitudes (except for polar lower stratosphere) to chemical control to upper altitudes. For example, in panels (d) in Figs. 8 and 10 (DJF), the ozone and temperature anomalies have opposite signs in the SH mid and high latitudes, which is considered to result from chemical control in the austral summer. In some regions of the NH high latitudes in DJF, the anomalies have the same sign, which is considered to result from dynamical control in boreal winter (the negative anomalies indicate less heat and ozone transport from the midlatitudes).

Fig. 11 shows the geopotential height (Z) anomalies associated with the QBO (QBOW-QBOE) at 10 hPa by a composite analysis. Positive geopotential height anomalies are significant in the tropics in all seasons. Zonally asymmetric Z anomalies associated with the QBO are evident and significant in the extratropics. During boreal winter (DJF), positive anomalies are evident in QBOW compared to QBOE from eastern North America to western Eurasia over the Arctic, indicating a weaker and more disturbed polar vortex. Over other regions of the Arctic, i.e., from eastern Eurasia to the North Pacific, there are negative geopotential height anomalies. This illustrates a shift of the polar vortex. While the polar vortex acts as a barrier that damps the meridional transport and mixing between the polar region and the midlatitudes, it is very cold and the ozone concentrations are very low in the polar vortex. This shift of the polar vortex therefore leads to positive ozone anomalies in eastern North America to western Eurasia and negative ozone anomalies in eastern Eurasia and the North Pacific (Fig. 8d). During boreal summer (JJA, winter in the SH), there are also evident zonally asymmetric anomalies of geopotential height associated with the QBO, with positive anomalies in the western hemisphere and negative anomalies in the eastern hemisphere over the Antarctic. However, ozone anomalies are all positive over the Antarctic from the merged satellite data (Fig. 8b). On the other hand, there are some significant negative ozone anomalies in the eastern hemisphere (0 °E to 140 °E) around the 60 °S from the ERA5 data (Fig. S6). Note that the periods of analysis in Fig. 8 (2002-2020) and Figs. 11 and S6 (1979-2020) are different because of the data availability, which may cause the different features of ozone anomalies associated with the QBO mentioned above. Whether the ozone QBO signals over the Antarctic are zonally asymmetric awaits further investigations. In boreal autumn (SON), zonally asymmetric geopotential height anomalies associated with the QBO are evident in both hemispheres. Positive Z anomalies are evident mainly in the North Atlantic and Eurasia over the Arctic during QBOW compared to QBOE, while negative anomalies are significant mainly in the North Pacific and North America. This leads to positive (weaker polar vortex)

and negative (stronger polar vortex) ozone anomalies in these regions, respectively (Fig. 8c). Over the Antarctic, geopotential height signals change in sign with negative anomalies from 60 °W to 180 °E and positive anomalies in other regions. Such changes of geopotential height associated with the QBO resemble the pattern of ozone anomalies, indicating the important role of the strength of the polar vortex in determining ozone concentrations.

To further illustrate the processes related to the geopotential height anomalies associated with QBO (QBOW-QBOE), we separate the monthly geopotential height anomalies into components of different wave numbers. Fig. 12 gives the overall changes of geopotential height as well as the corresponding changes in wave numbers 1-3 during QBOW over the NH in boreal winter (DJF). The climatological mean of the geopotential height during QBOE and its wave numbers 1-3 components are also shown. Comparing the signals in Fig. 12a and other figure panels, it is obvious that the geopotential height anomalies are dominated by the wave number 1 (wave-1) process. Relative to the climatological pattern (contour lines in Fig. 12b), the QBO related wave-1 anomalies (QBOW-QBOE) show an eastward phase shift by about 110 °, with more areas (about 60%) out of phase than in phase with it (Fig. 12b). For wave numbers 2-3, the QBO related anomalies are much weaker compared to wave-1 and shows destructive interfaces with the climatological waves (Figs. 12c-d).

The three dimensions T-N wave flux (Takaya and Nakamura, 2001) is employed to show the corresponding wave activity changes associated with the QBO. During QBOW, more planetary waves propagate upward over the eastern Eurasia and the North Pacific sector (60 °E to 120 °W, red contour lines in Fig. 13a) of the Arctic (north of 70 °N), which is consistent with the Holton-Tan mechanism (Holton and Tan, 1980). However, downward (or less upward) propagation of planetary waves is seen in other sectors of the Arctic (blue contour lines in Fig. 13a). The favourable upward propagation of planetary waves over eastern Eurasia and the North Pacific may be due to the relatively large climatological wave flux from the troposphere to the stratosphere in these regions (Elsbury et al., 2021). This leads to a weakening of the zonal wind in eastern Eurasia and the North Pacific but an enhancement of zonal wind in other sectors of the Arctic (north of 70 °N) due to the wave-mean flow interactions (shading in Fig. 13b). At the same time, more waves propagate poleward from lower latitudes to North America and the North Atlantic regions but equatorward from the North Pacific to lower latitudes (vectors Fig. 13a). Due to the wave-mean flow interactions, the convergence of the waves in North America and the North Atlantic, and the divergence of waves in eastern Eurasia and the North Pacific (shading in Fig. 13a) lead to deceleration and acceleration of zonal winds (shading in Fig. 13b) over these regions, respectively. On the other hand, the weakening (strengthening) of the zonal wind in the Arctic (north of 70 °N) also contributes to the stronger (weaker) westerlies in the subpolar regions (50-70° N) over eastern Eurasia and the North Pacific (other sectors) to conserve angular momentum and maintain mass continuity (Kidston et al., 2015). This indicates a shift of the polar vortex in the subpolar regions from North America and the North Atlantic to eastern Eurasia and the North Pacific, which can also be seen in the geopotential height anomalies as shown in Fig. 12a. This shift of the polar vortex is consistent with previous studies (e.g. Elsbury et al., 2021). As reported by previous studies, such a shift of polar vortex may cause downward wave propagation in the North America and the North Atlantic regions (Zhang et al., 2019; Elsbury et al., 2021), which is consistent with our results (blue contour lines in Fig. 13a).

The monthly geopotential height anomalies separated into components of different wave numbers are also discussed in SON in both the NH and SH hemispheres, since the zonally asymmetric features of QBO anomalies (QBOW-QBOE) are especially

significant. In boreal autumn, the QBO anomalies in geopotential height are also dominated by the wave number 1 process, and the wave-1 anomalies are out of phase with the climatological pattern (Fig. S8). For the SH in its spring (SON), the geopotential

height anomalies associated with the QBO are also dominated by the wave-1 process, but the wave-1 anomalies are shifted by about 90°compared to the climatological pattern (Fig. S9). These changes in geopotential heights and polar vortex can also be explained by the T-N wave flux activities (not shown). The phase shift or out of phase of wave activities compared to their climatological structure is a very complex issue and is out of the scope of this study. Anyway, it is clear that the QBO affects the polar vortex mainly through wave-1 process and leads to zonally asymmetric features in geopotential height and ozone

anomalies.

## 4  Conclusions and Discussion

The influences of the QBO on ozone have been studied for over 30 years. Here, we first revisited the influences of the QBO (QBOW-QBOE, using a QBO index at 20 hPa) on TCO and zonal mean stratospheric ozone using a longer record of merged satellite data (1979-2020 for TCO and 1984-2020 for zonal mean ozone), together with the most recent ERA5 reanalysis and

NCAR's CESM-WACCM model simulations. We found seasonally independent positive ozone anomalies (QBOW-QBOE) in the tropics and negative ozone anomalies during winter-spring of the respective hemisphere in the extratropics, which confirms the results from previous studies (Bowman, 1989; Hamilton, 1989; Tung and Yang, 1994; Randel and Wu, 1996; Butchart et al., 2003). The ERA5 data and the CESM-WACCM model capture the QBO signals in ozone very well, and a sensitivity simulation without QBO nudging further confirms the robustness of the QBO impacts on ozone. Some new features in TCO corresponding

to the QBO are also found from the global distribution pattern: zonally asymmetric TCO anomalies are evident during autumn-winter in the NH and during spring in the SH over the polar regions; positive TCO anomalies are seen from North America to the North Atlantic (120°W-30°E) over the Arctic in winter (DJF) and mainly in Greenland to Eurasia (60°W-120°E) over the Arctic in autumn (SON); Some positive signals in TCO exist during its spring (SON) in the South America sector over the Antarctic. The TCO anomalies are contributed both by the ozone changes in the lower and middle stratosphere, which are

mainly caused by the vertical transport related to the vertical gradient of climatological ozone distribution and changes in the vertical component of the BDC.

We then further investigated the global distribution of ozone anomalies related to the QBO (QBOW-QBOE) in the strato-sphere in monthly mean anomalies as well as in different seasons. Similar to the TCO anomalies described above, evident zonally asymmetric features can be found in ozone anomalies in the middle and lower stratosphere associated with QBO,

which are especially significant during autumn-winter (SON-DJF) over the Arctic and during spring (SON) over the Antarctic. The zonally asymmetric features of ozone associated with the QBO in the middle (at ∼10 hPa) and lower (at ∼50 hPa) strato-sphere are in general consistent with the spatial pattern of TCO as described above, indicating the dominant contribution of the stratospheric ozone to the TCO variations. According to the analysis of meteorological parameters, we found that the QBO influences on ozone are mainly through dynamical transport and also related to temperature-dependent chemical production.

Besides the well-known weakening/strengthening of the polar vortex during the easterly/westerly phase of the QBO (using 50

hPa U) (Anstey and Shepherd, 2014; Watson and Gray, 2014; Andrews et al., 2019; Xie et al., 2020), the QBO (using PC1 close to 20 hPa U as introduced in Methods) leads to a wave number 1 pattern in geopotential height anomalies over the polar regions in boreal winter. Such wave-1 geopotential height anomalies can be explained by the zonal asymmetric changes of wave activities as shown by the T-N flux. The wave-1 anomalies are shifted eastward by about 110 °during boreal winter (DJF) in the NH, out of phase during autumn in the NH, and shifted by about 90°during spring (SON) in the SH compared to the climatological wave-1 pattern.

Note that the QBO index used in this study is equivalent to the equatorial zonal winds (U) at 20 hPa, which is different with the more widely used 50 hPa U index. Figure S10 also shows the TCO anomalies between different QBO phases in different seasons using the PC2 (indicating the equatorial zonal winds at 50 hPa) as the QBO index. In general, there are also some zonally asymmetric features in the differences of TCO between QBOW and QBOE phases and the magnitude of the anomalies are comparable to that shown in Figure 3. In DJF, the QBO signals depending on PC2 are opposite in sign with that on PC1 in the Northern Hemisphere (NH), while the zonal asymmetry is not that obvious. In MAM, the TCO anomalies are all negative over most of areas in the mid-to-high latitudes of the NH no matter which QBO index is used. In JJA, the PC2 related TCO anomalies are in the same sign with the PC1 related anomalies in the NH, but opposite in sign with the PC1 related anomalies in the Tropics and the Southern Hemisphere (SH). In SON, the zonal asymmetry of the PC2 related TCO anomalies is more obvious in the SH but less significant in the NH compared with PC1. It is very interesting that there are significant differences in the QBO related signals while using QBO index at different levels, e.g, the PC2 related TCO anomalies are more zonal asymmetry during SON in the SH but less zonal asymmetry during DJF in the NH. The exact reason for these differences is out of the scope of this study and awaits further studies.

Stratospheric ozone is not only essential in protecting life on the Earth, but also has important climate impacts on the surface. More and more studies reported the important role of ozone variations in modifying the stratospheric circulation and therefore influencing the surface climate (e.g. Xie et al., 2020). Since the QBO has relatively high predictability, considering its impacts on stratospheric ozone and subsequent atmospheric circulations may be helpful to improve the prediction of surface weather and climate.

*Data availability.* The MSR2, merged satellite data and the ERA5 reanalysis can be downloaded from the Copernicus Climate Change Service website https://cds.climate.copernicus.eu/cdsapp#!/home. The model simulations can be provided to readers by contacting the corresponding author.

*Author contributions.* W. W. performed the data analysis, plotted the figures and wrote the first draft of the paper. J. H., M. S., H. W. and S. Z. contributed to the interpretation of the results and revised the manuscript. W. J. contributed to the wave number analysis. All authors reviewed the manuscript.

*Competing interests.* The authors declare that they have no conflict of interest.

*Acknowledgements.* This work was supported by the Ministry of Science and Technology of the People's Republic of China (2019YFE0125000), National Natural Science Foundation of China (Grant No. 42075055 and 41875095) and the open project (KLME202003) from the Key Laboratory of Meteorological Disaster (KLME), Ministry of Education & Collaborative Innovation Center on Forecast and Evaluation of Meteorological Disasters(CIC-FEMD). We thank the Copernicus Climate Change Service for providing the satellite data, ECMWF for the ERA5 data and NCAR for the CESM-WACCM Model. The model simulations were carried out at the Deutsche Klimarechenzentrum (DKRZ) in Hamburg, by Prof. Katja Matthes's group at the Helmholtz Centre for Ocean Research Kiel (GEOMAR), Germany. We thank the two anonymous reviewers and the editor for the very helpful comments and suggestions.

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

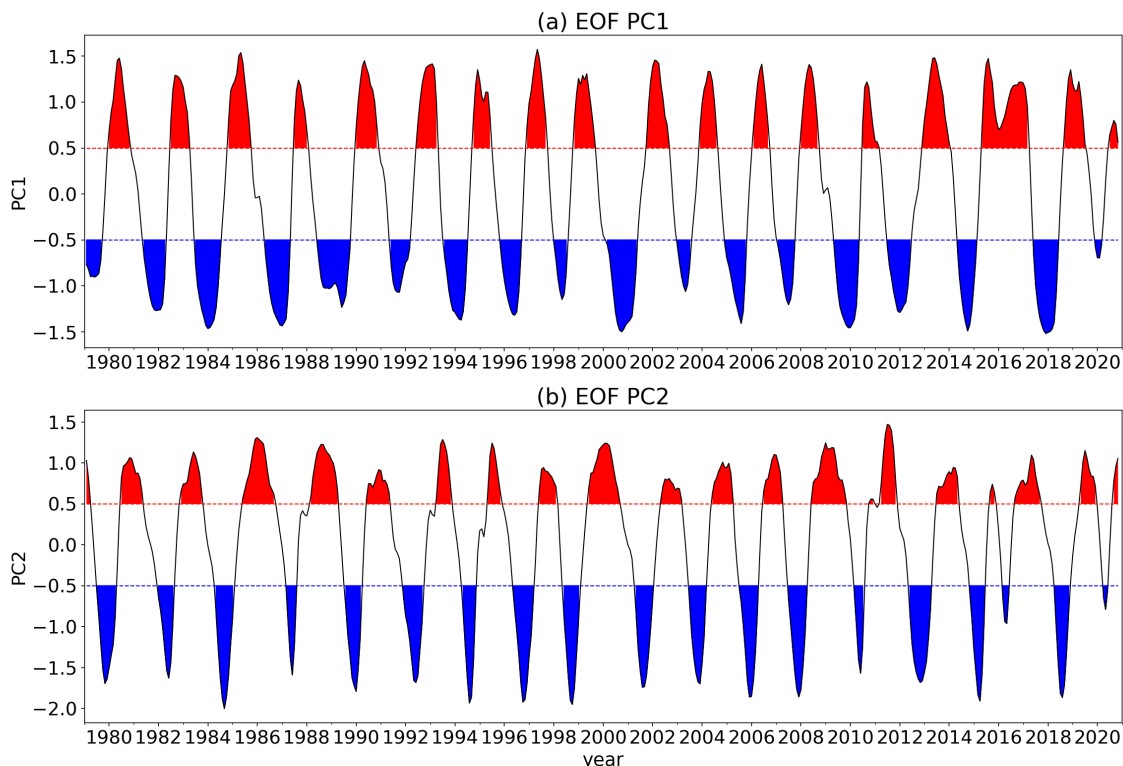

**Figure 1.** Time series of the first two principal components (PCs) of QBO from 1979 to 2020 obtained by an EOF analysis applied to the FUB QBO data from 70 to 10 hPa. **(a)** PC1, indicating the QBO phase in the middle stratosphere at  20 hPa. **(b)** PC2, indicating the QBO phase in the lower stratosphere at  50 hPa. Red/blue colours are shaded where the QBO indexes are greater/less than half of its standard deviation, indicating the QBOW/QBOE phases.

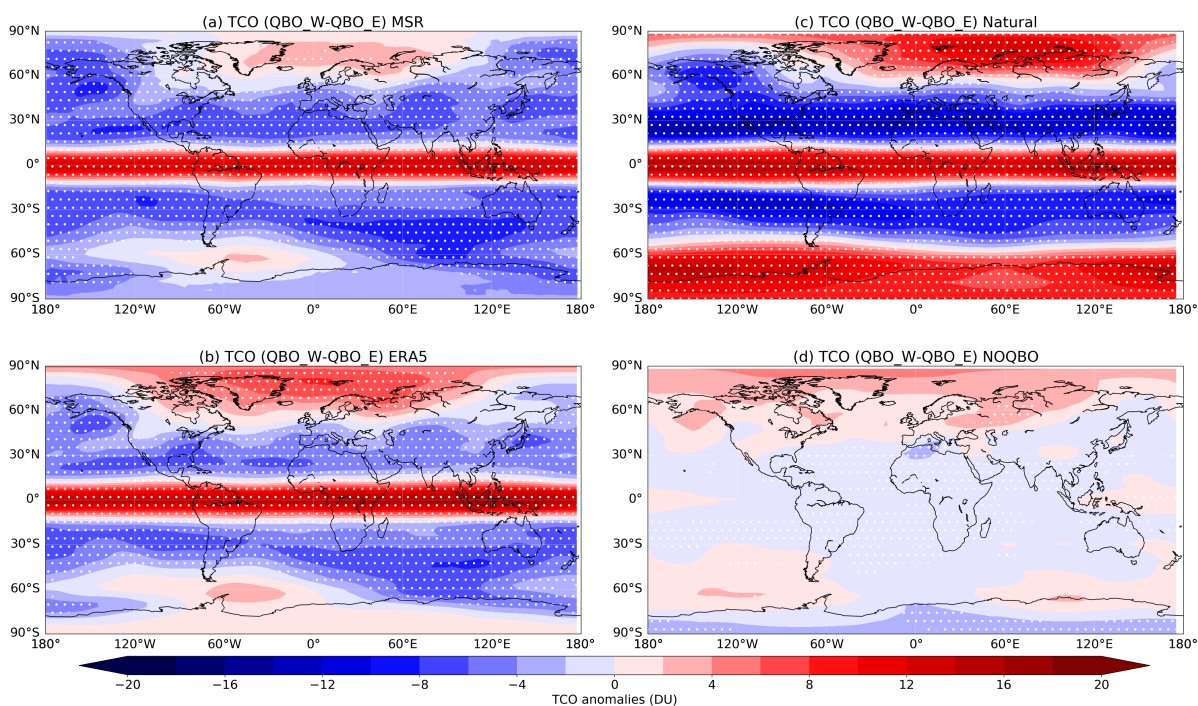

**Figure 2.** Influences of QBO (QBOW-QBOE) on global total column ozone (TCO) based on monthly anomalies from different data sets. **(a)** MSR2 data 1979-2020. **(b)** ERA5 data 1979-2020. **(c)** CESM-WACCM Natural run 1979-2020. **(d)** CESM-WACCM NOQBO run 1979-2020. Stippled areas indicate results that are statistically significant over the 95% level, using the two-tailed Student's t-test.

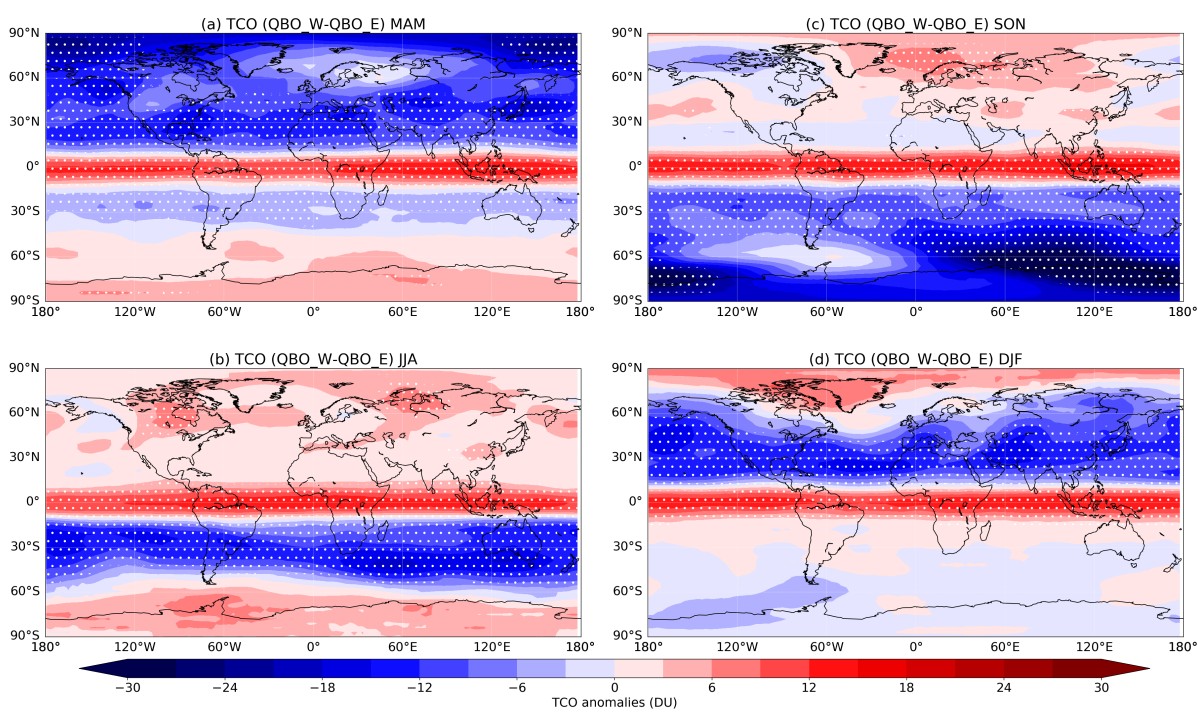

**Figure 3.** Influences of QBO (QBOW-QBOE) on global total column ozone (TCO) in different seasons based on MSR2 data 1979-2020. **(a)** MAM. **(b)** JJA. **(c)** SON. **(d)** DJF. Stippled areas indicate results that are statistically significant over the 95% level, using the two-tailed Student's t-test.

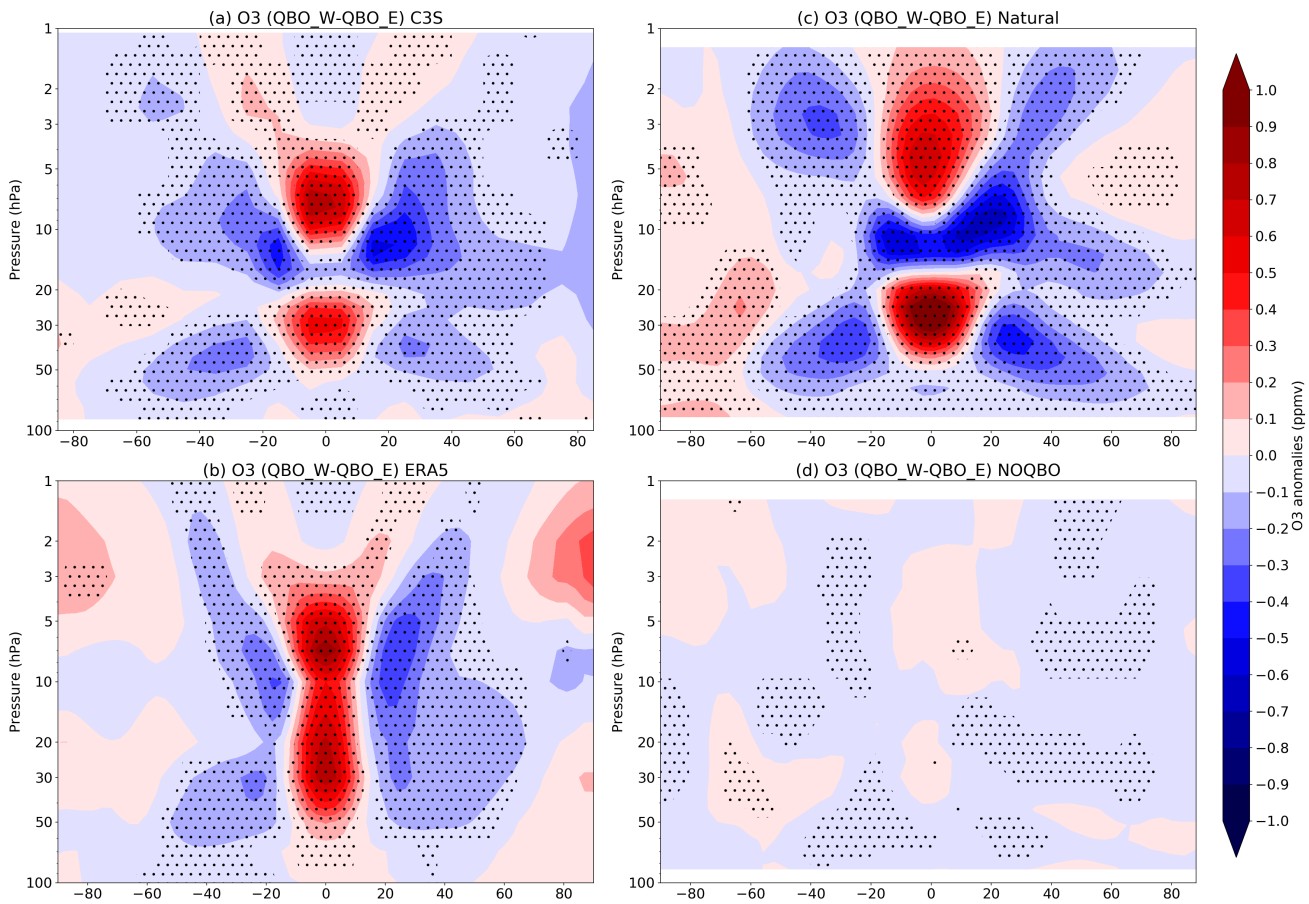

**Figure 4.** Latitude-height cross-section of ozone anomalies associated with QBO (QBOW-QBOE) based on monthly anomalies of zonal mean ozone from different data sets for the period 1985-2020. **(a)** Merged satellite data from C3S. **(b)** ERA5 data. **(c)** CESM-WACCM Natural run. **(d)** CESM-WACCM NOQBO run. Stippled areas indicate results that are statistically significant over the 99% level, using the two-tailed Student's t-test.

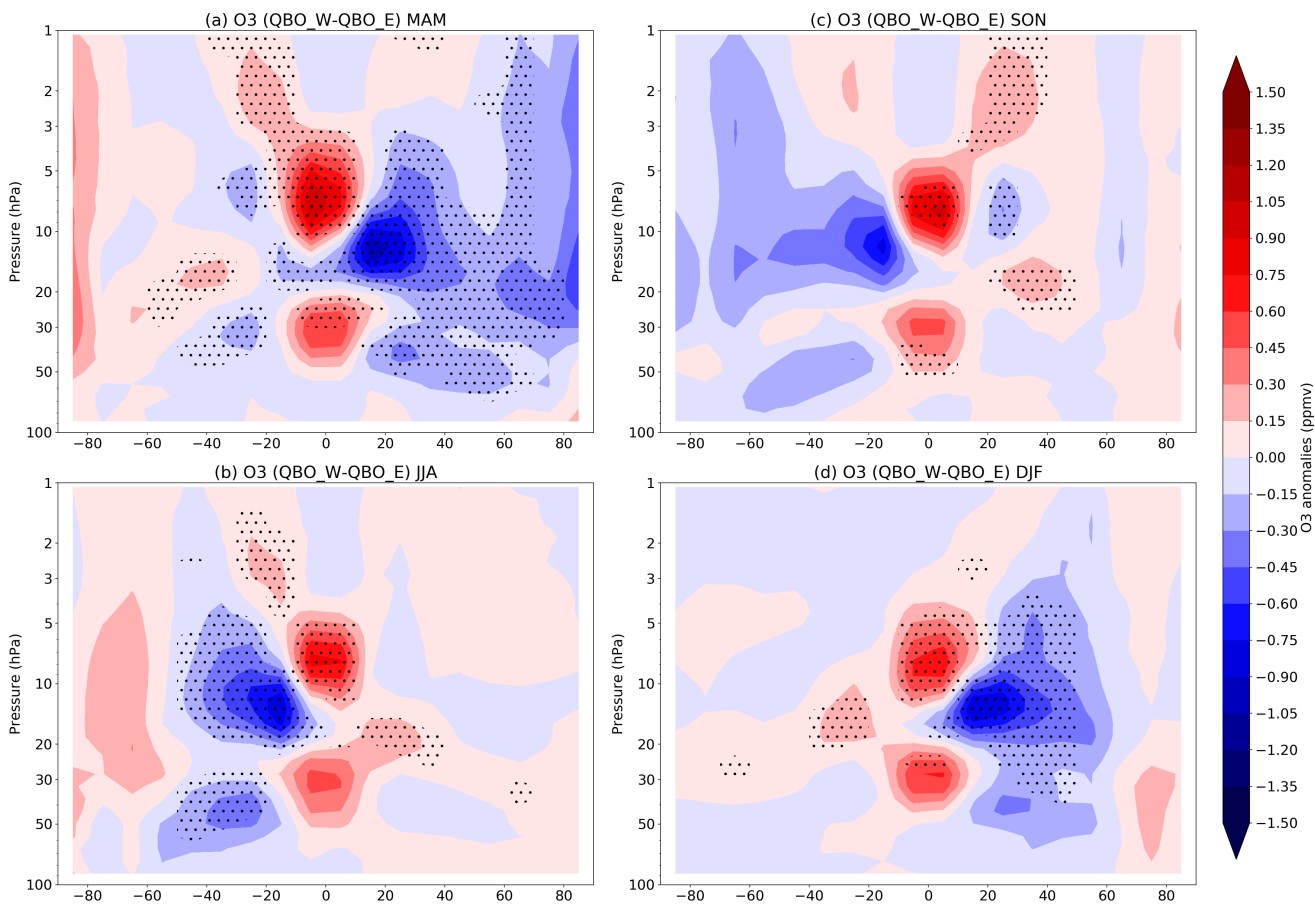

**Figure 5.** Latitude-height cross-section of ozone anomalies associated with QBO (QBOW-QBOE) based on merged satellite data from C3S for the period 1985-2020. **(a)** MAM. **(b)** JJA. **(c)** SON. **(d)** DJF. Stippled areas indicate results that are statistically significant over the 95% level, using the two-tailed Student's t-test.

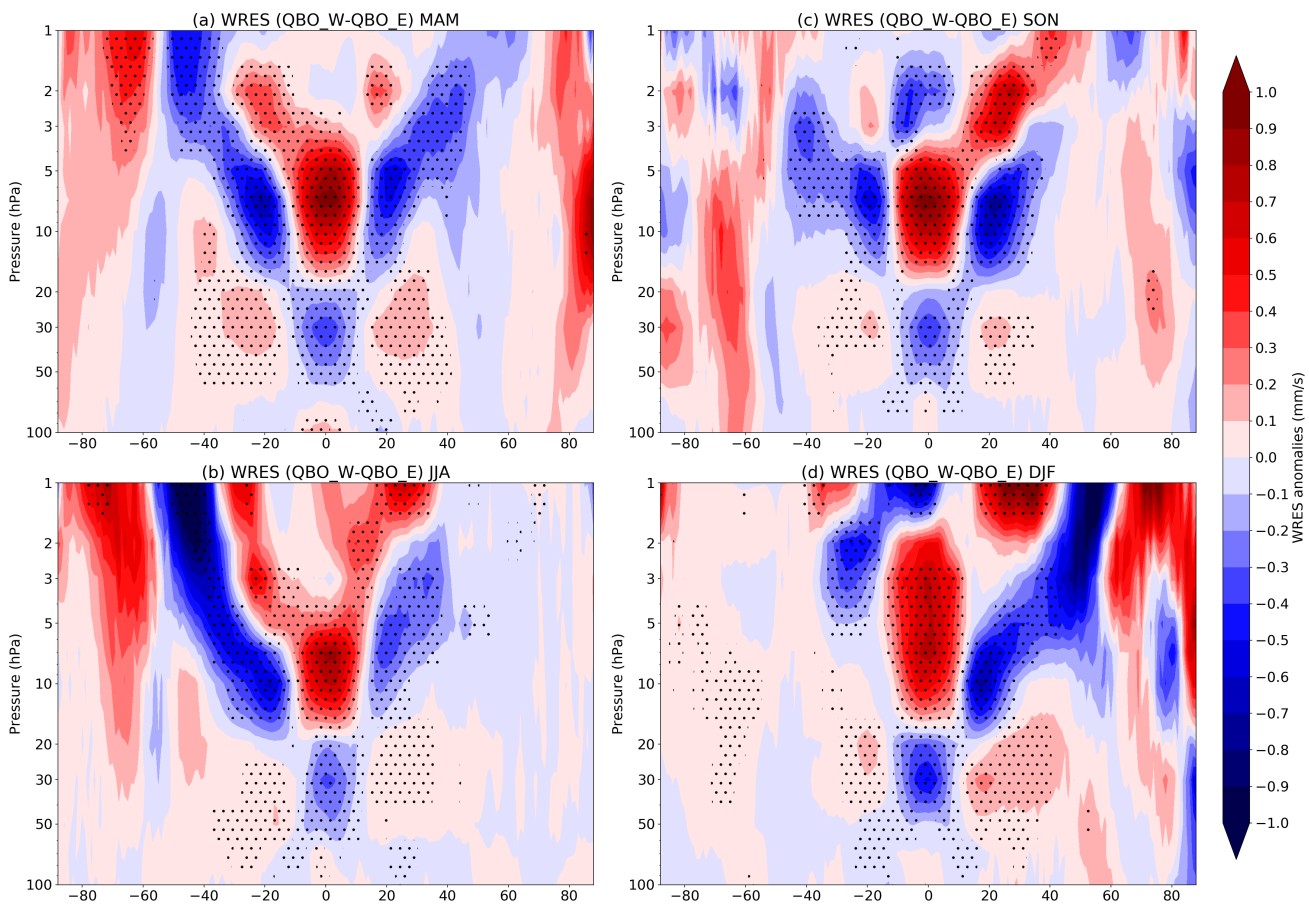

**Figure 6.** Latitude-height cross-section of w* (vertical component of the BDC) anomalies associated with QBO (QBOW-QBOE) from ERA5 data for the period 1985-2020. **(a)** MAM. **(b)** JJA. **(c)** SON. **(d)** DJF. Stippled areas indicate results that are statistically significant over the 95% level, using the two-tailed Student's t-test.

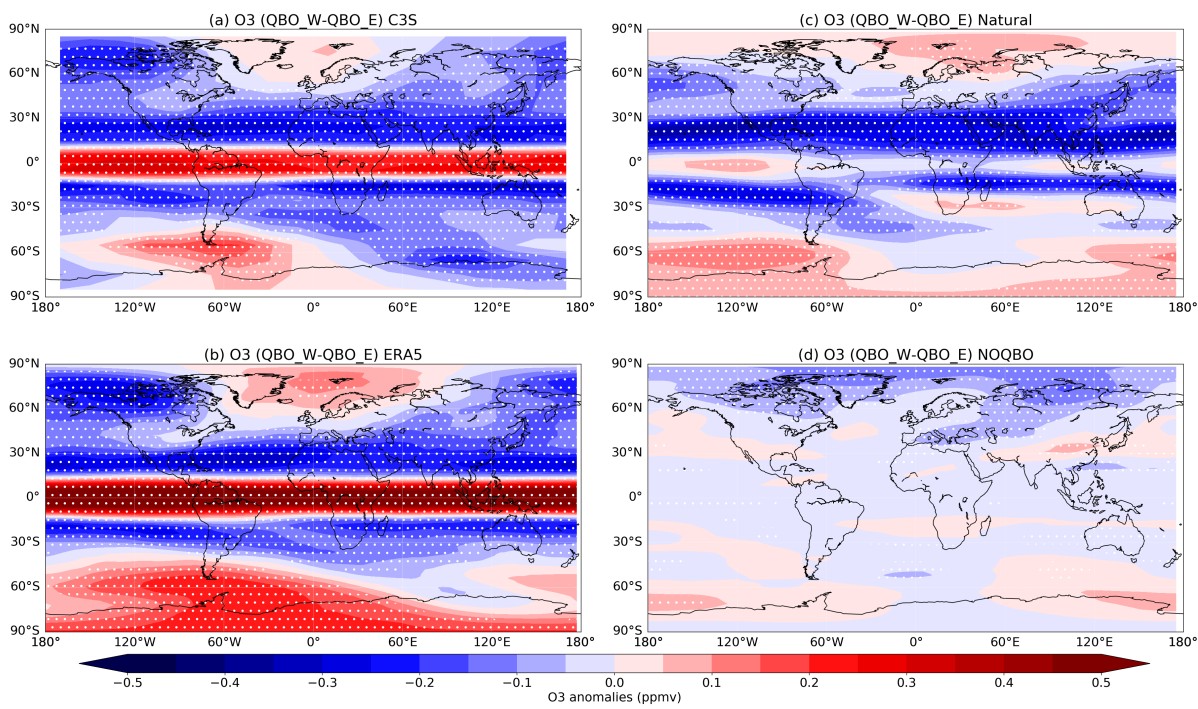

**Figure 7.** Influences of QBO (QBOW-QBOE) on global stratospheric ozone (10 hPa) based on monthly anomalies from different data sets for the period 2002-2020. **(a)** Merged satellite data from C3S. **(b)** ERA5 data. **(c)** CESM-WACCM Natural run. **(d)** CESM-WACCM NOQBO run. Stippled areas indicate results that are statistically significant over the 95% level, using the two-tailed Student's t-test.

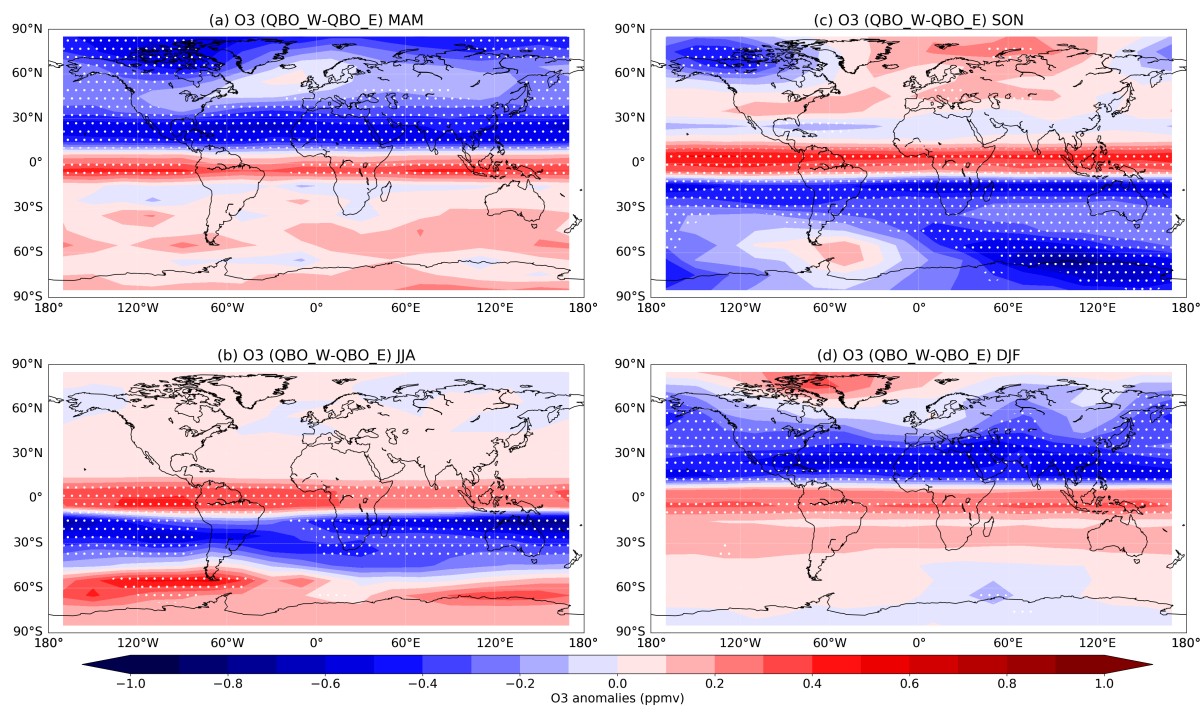

**Figure 8.** Influences of QBO (QBOW-QBOE) on global stratospheric ozone (10 hPa) based on merged satellite data from C3S for the period 2002-2020. **(a)** MAM. **(b)** JJA. **(c)** SON. **(d)** DJF. Stippled areas indicate results that are statistically significant over the 95% level, using the two-tailed Student's t-test.

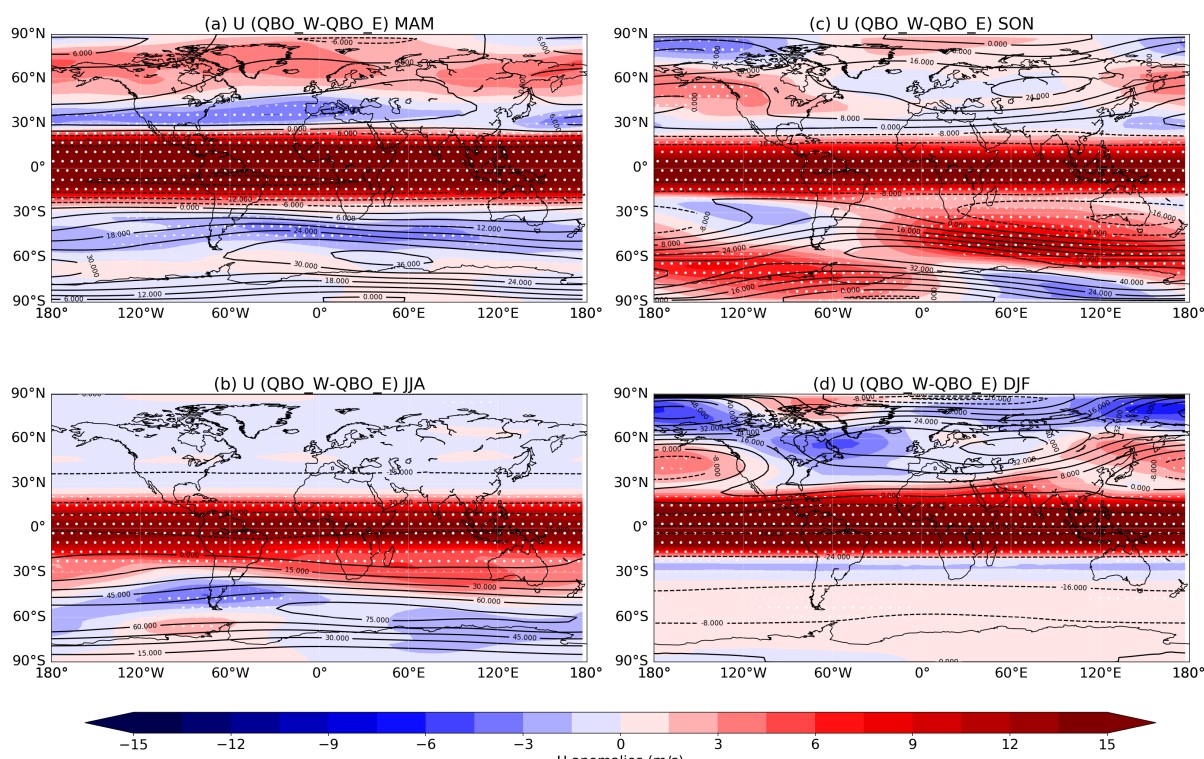

**Figure 9.** Influences of QBO (QBOW-QBOE) on the global zonal wind (U at 10 hPa) based on ERA5 data for the period 1979-2020. The climatological values of zonal winds during QBOE in each season are also shown (contour lines, with contour intervals of 6, 15, 8 and 8 m/s in **(a)**, **(b)**, **(c)**, and **(d)**, respectively). **(a)** MAM. **(b)** JJA. **(c)** SON. **(d)** DJF. Stippled areas indicate results that are statistically significant over the 95% level, using the two-tailed Student's t-test.

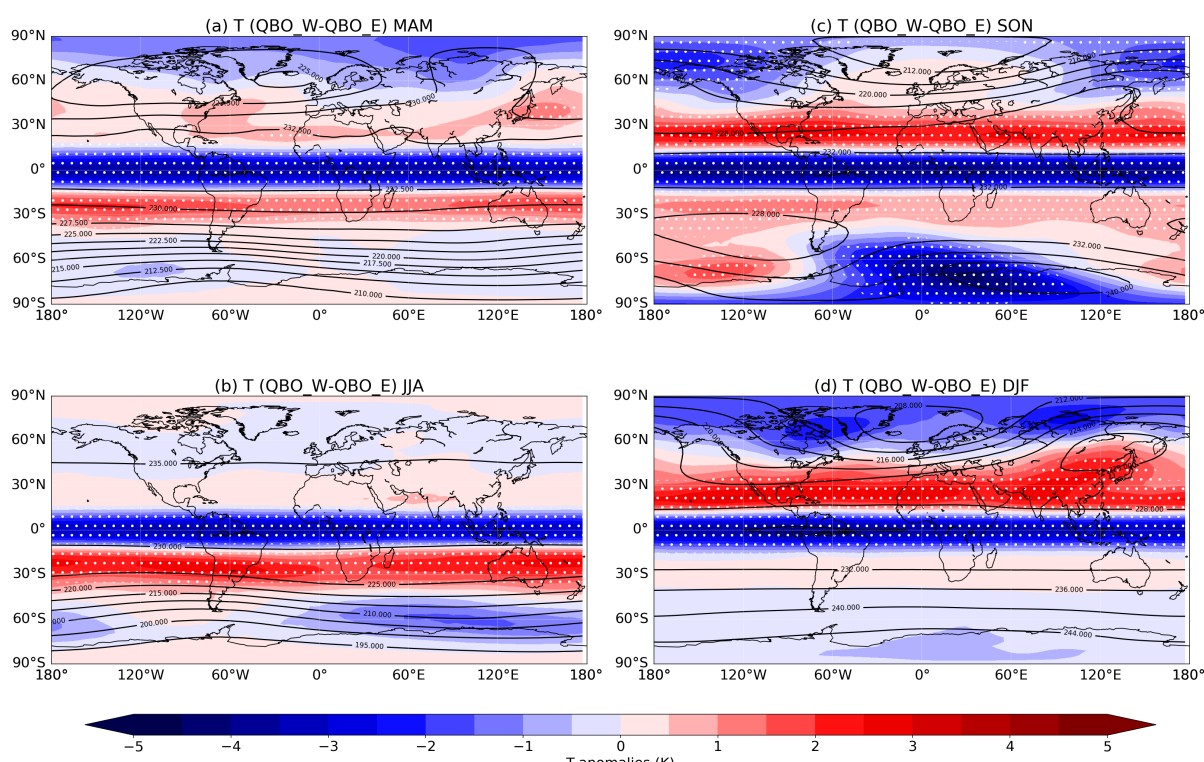

**Figure 10.** Influences of QBO (QBOW-QBOE) on global temperature (T at 10 hPa) based on ERA5 data for the period 1979-2020. The climatological values of temperature during QBOE in each season are also shown (contour lines, with contour intervals of 2, 5, 4 and 4 K in (**a**), (**b**), (**c**), and (**d**), respectively). (**a**) MAM. (**b**) JJA. (**c**) SON. (**d**) DJF. Stippled areas indicate results that are statistically significant over the 95% level, using the two-tailed Student's t-test.

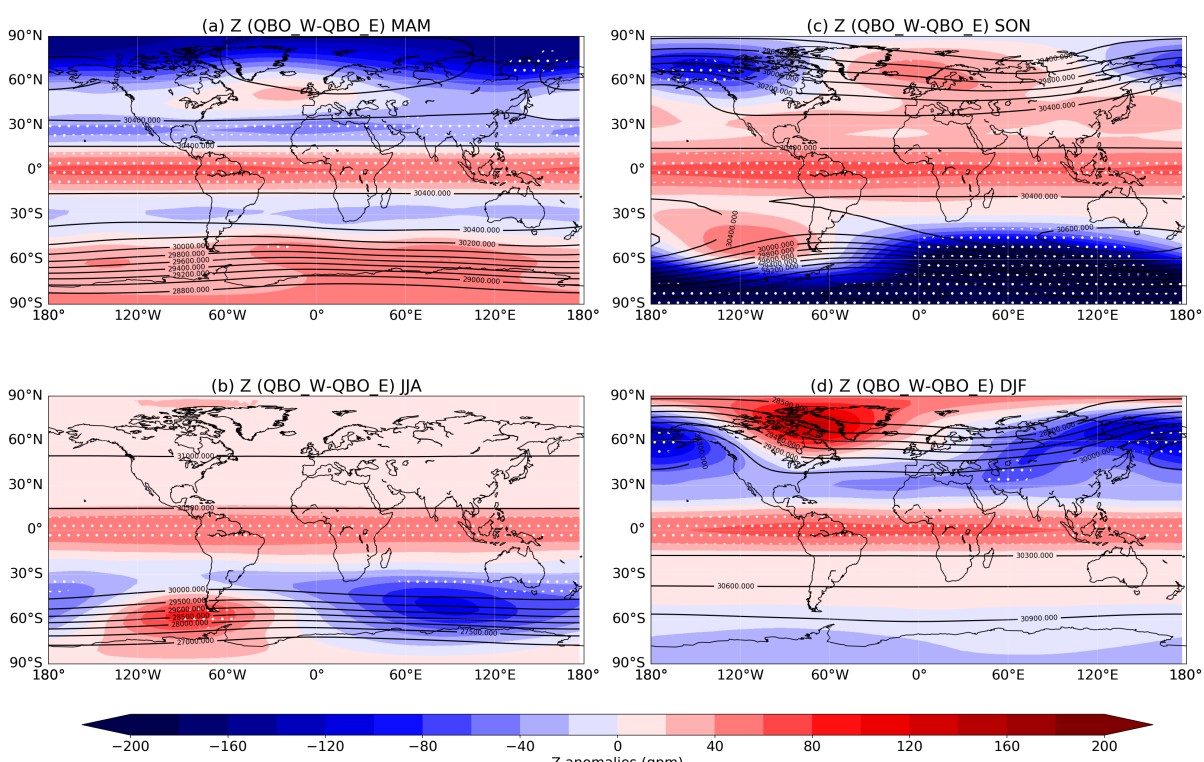

**Figure 11.** Influences of QBO (QBOW-QBOE) on global geopotential height (Z at 10 hPa) based on ERA5 data for the period 1979-2020. The climatological values of geopotential height during QBOE in each season are also shown (contour lines, with contour intervals of 200, 500, 200 and 300 gpm in **(a)**, **(b)**, **(c)**, and **(d)**, respectively). **(a)** MAM. **(b)** JJA. **(c)** SON. **(d)** DJF. Stippled areas indicate results that are statistically significant over the 95% level, using the two-tailed Student's t-test.

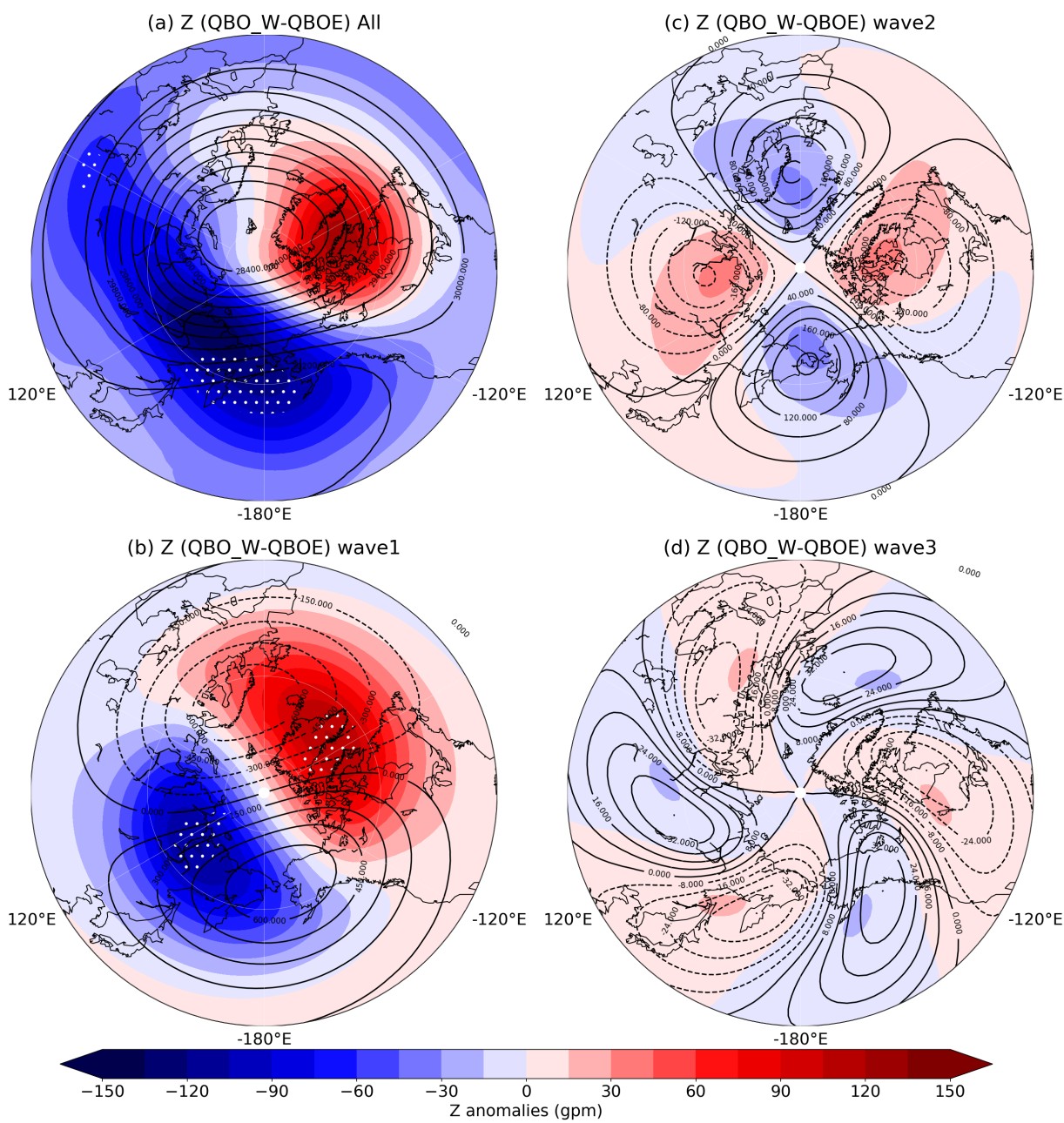

**Figure 12. (a)** Influences of QBO (QBOW-QBOE) on geopotential height (Z at 10 hPa) in the northern hemisphere winter (DJF) based on ERA5 data for the period 1979-2020. **(b-d)** The corresponding changes of geopotential height associated with QBO in wave numbers 1-3. The climatological values of geopotential height in winter as well as the climatological patterns of wave numbers 1-3 during QBOE are also shown (contour lines, with contour intervals of 200, 150, 40 and 8 gpm in **(a)**, **(b)**, **(c)**, and **(d)**, respectively). Stippled areas indicate results that are statistically significant over the 95% level, using the two-tailed Student's t-test.

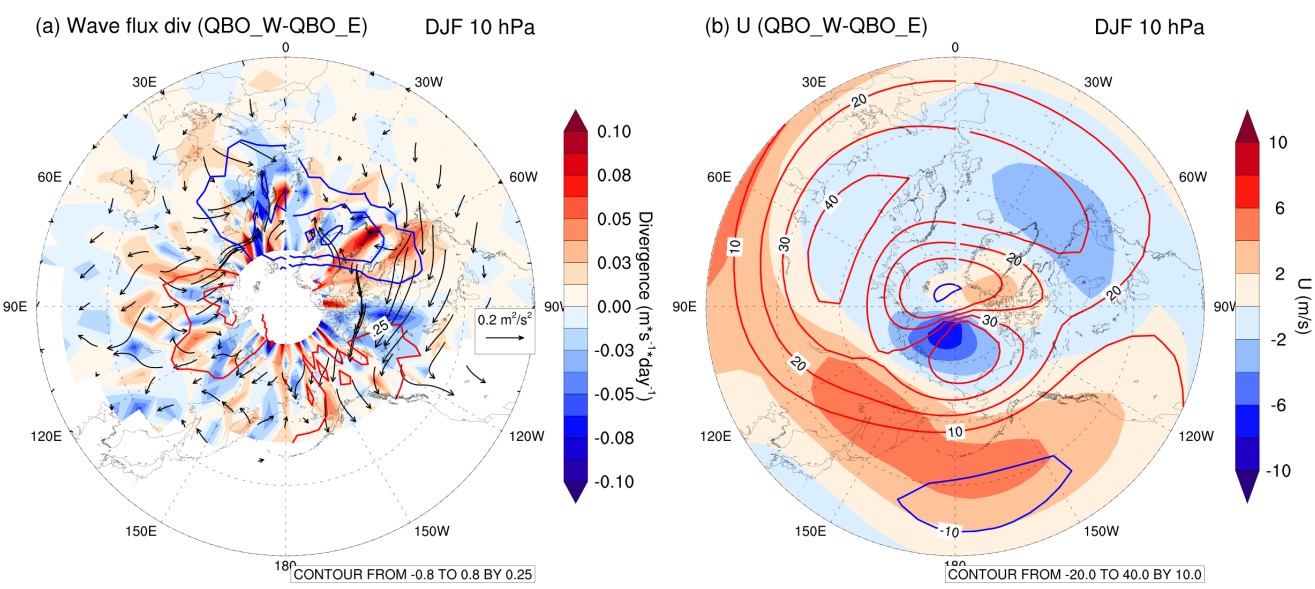

**Figure 13.** Influences of QBO (QBOW-QBOE) on T-N wave flux **(a)** and zonal winds **(b)** at 10 hPa north of 30 °N during winter (DJF) based on ERA5 data for the period 1979-2020. In **(a)**, the meridional and zonal components of the wave flux are shown as vectors, the vertical component is shown as contour lines (positive in red) and the divergence of the wave flux is shaded. In **(b)**, the climatological values of zonal wind during QBOE are shown in contour lines (solid lines for westerly, with a contour interval of 10 m/s) and the anomalies are shaded. Stippled areas indicate results that are statistically significant over the 95% level, using the two-tailed Student's t-test.