# Peer review of "Zonally Asymmetric Influences of the Quasi-Biennial Oscillation on Stratospheric Ozone"

_Atmospheric Chemistry and Physics, 2022_

## Author Response (AR1)

**Responses to Reviewer 1**

Comments on "Zonally asymmetric influences of the Quasi-Biennial Oscillation on stratospheric ozone" by Wang et al.

General comments

This paper reports a global ozone anomaly and associated meteorological field anomalies due to the QBO. Merged satellite data of the ozone and its column amount, ERA5 reanalysis data, and CESM-WACCM model simulation output are used for analysis. The authors analyzed the difference in ozone and meteorological fields between the westerly and easterly phase composites and showed the QBO signals globally. In particular, the signals at high latitudes showed a clear zonal asymmetry. The authors also discuss seasonal differences in the QBO signals and their zonal asymmetry.

I think the results presented in this manuscript are interesting and scientifically valuable. However, I would like to recommend carefully and thoughtfully describing the correspondence of their results to those in preceding studies that were performed during shorter period and reported as a function of latitude. This would help this research be more valuable in the research field. Moreover, there are some misleading descriptions of chemical effects on ozone anomalies in the tropical middle and upper stratosphere. Therefore, I recommend that some revisions be made before acceptance.

We thank the reviewer for the very helpful comments. We have revised the manuscript carefully based on the comments and suggestions of the reviewer and hope that the manuscript has been improved significantly. More details of the revision can be found in the revised manuscript as well as the point-to-point response as follows. The comments are shown in black and our replies are marked as blue.

Major comments

As I stated in the general comments, I think that more carefully describing the correspondence of this study's results to results from preceding studies reported as a function of latitude (wave amplitude, zonal-mean zonal winds, temperature, etc.) may greatly improve this paper scientifically. The analysis of the zonal asymmetry of QBO signals is new and interesting. However, preceding studies also imply zonal asymmetry through the wave amplitude or wave flux (E-P flux). For example, Holton and Tan (1980) suggested that the wave amplitude in the high-latitude stratosphere may change depending on the QBO phase. This already indicates a change in the zonal asymmetry of the dynamical field and in the strength of the zonal-mean zonal wind. Figure 12 is an interesting figure that demonstrates the longitudinal phase of the QBO signals and less zonal asymmetry of the geopotential height field in the westerly phase of QBO as compared to the easterly phase using climatology (contours) and anomaly (colors)

fields, with a slight phase shift from the climatology of wave number one, which is the dominant mode of the wave activity. I would suggest that the authors explain the connection of the 3D anomalies due to the QBO to the zonal-mean anomalies as a function of latitude.

We thank the reviewer very much for the constructive suggestion. We have read through more literatures and added further analysis using the wave flux (T-N Flux, Takaya and Nakamura, 2001). Now we have some discussion about the connection between the zonal-mean anomalies and the zonal asymmetric features. As reported by previous literatures, during QBOW at 20 hPa (QBOE at 50 hPa), there are enhanced upward wave fluxes from the troposphere to the stratosphere in high-latitudes of the northern hemisphere in DJF (e.g., Naoe and Shibata 2010; Elsbury et al., 2020). However, the planetary waves propagate upward preferred over the regions of eastern Eurasia and north America (Figure R1a and also in Figures S5f and S5h of Elsbury et al., 2020, note that what they show are anomalies during QBOE while we show in QBOW), maybe due to the large climatological planetary wave flux in this sector (White et al., 2019). At the same time, seen from its meridional and zonal components, the T-N Flux converges over the north Atlantic sector but diverges over the north Pacific sector in the high-latitudes, which leads to acceleration and deceleration of zonal winds, respectively (Figure R1b). Such asymmetric wave propagation leads to perturbations of the polar vortex, i.e. a trough over the eastern Eurasia and North Pacific sector and a ridge over the North Atlantic sector (Figure R1c). The shift of the polar vortex from the North Atlantic to the Eurasia and North Pacific sector results in downward propagation of planetary waves over the North Atlantic (Figure R1a, and also in Zhang et al., 2019; Elsbury et al., 2020).

[Figure]

[Figure]

Figure R1. Influences of QBO (QBOW-QBOE) on T-N Flux (a), zonal wind (b) and geopotential height (c) in different seasons at 10 hPa based on ERA5 data for the period 1979-2020. The meridional and zonal components of T-N Flux are shown as vectors and the vertical component is shaded in (a). In (b) and (c), the climatological mean is also shown as contour lines and the QBO related anomalies are shaded. Stippled areas indicate results that are statistically significant over the 95% level, using the two-tailed Student's t-test.

Another point is that the author should state the chemical effect on the ozone anomaly in the middle and upper stratosphere. To clarify the chemical effect in the QBO, I recommend that the authors show a latitude–height cross section of the temperature anomaly, such as in Figs. 5 and 6, and discuss the possibility of a chemical effect. As shown in Fig.6, positive anomalies of w* are evident above the ozone mixing ratio maximum (around 10 hPa), and accordingly, positive ozone anomalies are also evident, as shown in Fig. 5. The authors said that this positive ozone anomaly was caused by transport above the ozone mixing ratio peak. However, I think that the ozone at these altitudes in the tropics is also influenced by chemistry (e.g., Fig.1 of Solomon et al., 1985). If temperature at these altitudes has negative anomalies associated with the positive anomalies of w*, then the chemical effect should lead to a positive ozone anomaly, because a lower temperature leads to more ozone due to the temperature dependence of reaction coefficients in the gas phase chemistry. Then the positive ozone anomaly is consistent with the chemically induced anomaly as well as the dynamically induced (transport) anomaly.

We thank the reviewer for the very helpful comments. We have added a figure in the supplemental material to show the latitude–height cross section of the temperature anomaly associated with QBO as the reviewer suggested. We agree with the reviewer that the temperature dependent chemical effects should also be considered. As the reviewer expected, negative temperature anomalies can be found in the middle stratosphere in the tropics, which contributes to the positive ozone anomalies correspondingly. We have added some discussion about this effect in the revised manuscript.

[Figure]

Figure R2. Latitude-height cross-section of temperature anomalies associated with QBO (QBOW-QBOE) based on ERA5 data for the period 1985-2020. **(a)** MAM. **(b)** JJA. **(c)** SON. **(d)** DJF. Stippled areas indicate results that are statistically significant over the 95% level, using the two-tailed Student's t-test.

Finally, the color range around the zero value is indicated by white in the most of the figures. This makes the positive and negative anomalies around zero hard to distinguish. It would be better to change the color scale so that the blue shades can indicate negative anomalies and the red shades can indicate positive ones, with the boundary at the zero value.

Thanks for the good comments. We have adapted all the figures as suggested.

Minor comments

Lines 24 and 25: "Fahey et al., 2018" should be "WMO, 2018"
Corrected.
Lines 145–147: The explanation of positive and negative anomalies around the South Pole is not evident from Figure 2(a) and (b) because the negative and positive anomalies are represented by the same color (white) in the range [-2, 2].
We have adapted Figure 2 and now it is more evident. Some of the descriptions are also modified due to the new figure.
Lines 175–176: The positive anomaly over the equator from ERA5 is not separated vertically, which is different from C3S.

Sorry for the inaccuracy description. We have updated the description as follows: "QBO signals in ERA5 ozone (Fig. 4b) are in good agreement with the merged satellite data, except that the positive anomalies over the equator from ERA5 are not separated vertically."

Lines 177–178: The positive anomaly in the upper stratosphere from the CESM-WACCM Natural run is located at a little higher altitude and extended higher than the observations.

Sorry for the inaccuracy description. We have updated the description as follows: "The CESM-WACCM model also shows good consistency with the satellite and ERA5 data in the Natural run with a QBO nudging (Fig. 4c), although the positive anomaly in the tropical upper stratosphere from the Natural run is located at a little higher altitude and extended higher than the observations, and the negative signals are extended higher up to the upper stratosphere in the extra-tropics.".

Lines 188–192: The transport effect is important in the lower stratosphere, but I think in the middle and upper stratosphere in the tropics, the chemical effect through temperature change is also important (e.g., Fig.1 of Solomon et al., 1985). For example, the positive ozone anomalies above 10 hPa in the tropics may partly or almost totally be caused by negative temperature anomalies that can be caused by the positive w* anomalies. It would be helpful if the authors could show the latitude–height cross section of temperature anomalies.

Thanks for the very helpful suggestion. We have added a figure to show the latitude–height cross section of the temperature anomaly associated with QBO as the reviewer suggested and discussed this in the revised manuscript.

Lines 207–208: If you discuss correspondence to TCO, checking the ozone anomaly around 50 hPa as well as 10 hPa would be necessary, because ozone concentration (molecules per volume) is at its maximum around 50 hPa. Although the anomaly at 50 hPa is described at the end of the paragraph, I would recommend mentioning ozone anomalies at these two pressure levels accordingly.

Thanks for the good comment. We have added two figures to show the corresponding changes of ozone at 50 hPa associated with QBO and also more discussion about the 50 hPa ozone anomalies in the revised manuscript.

Lines 209–211: What is the meteorological field behind this ozone anomaly distribution at 10 hPa? Are Figures S5 and S6 helpful to explain it?

Yes, the ozone anomalies can be explained by the geopotential height anomalies as shown in Fig. R3c and Figs. S5-S6. Comparing Figure R3 and Figure 8 in the main text, positive ozone anomalies are well located in the regions with positive geopotential height anomalies, which indicate a weaker polar vortex. We have added some discussion about this in Section 3.4.

[Figure]

**Figure R3.** Influences of QBO (QBOW-QBOE) on global geopotential height (Z at 10 hPa) based on ERA5 data for the period 1979-2020. The climatological mean of geopotential height in each season is also shown (contour lines). (a) MAM. (b) JJA. (c) SON. (d) DJF. Stippled areas indicate results that are statistically significant over the 95% level, using the two-tailed Student's t-test.

Lines 239–240: I do not agree. In the framework of gas phase chemistry, a low-temperature anomaly leads to a high ozone-concentration anomaly due to the temperature dependence of reaction coefficients. The region where the low-temperature anomaly leads chemically to a low-ozone anomaly is limited in the polar lower stratosphere where heterogeneous reactions on the PSCs work.

We apologize for the mistake here. Yes, low temperatures should lead to high ozone concentrations in the tropics of the stratosphere. We have corrected the description correspondingly.

Line 250: I think that over the Antarctic, the ERA5 data show negative anomalies in the western hemisphere as well as the eastern hemisphere. A zonally asymmetric anomaly is evident only around 60ºS.

Sorry for the inaccuracy description. We have updated the description as follows:

"On the other hand, there are some negative ozone anomalies in the eastern hemisphere (0-140 º W) around 60º S from the ERA5 data (Fig. S3), although the signals are not statistically significant."

Lines 293–294: I do not agree in terms of ozone in the middle and upper stratosphere in the topics but agree in terms of TCO.

Sorry for the inaccuracy description. We have updated the description as follows:

"According to the analysis of meteorological parameters, we found that the QBO influences on ozone are related to both dynamical transport and temperature-dependent chemical production.".

References:

Elsbury, D., Peings, Y., and Magnusdottir, G.: Variation in the Holton-Tan effect by longitude, Quart. J. Roy. Meteor. Soc., 147, 1767–1787, https://doi.org/10.1002/qj.3993, 2021.

Garfinkel, C. I., Shaw, T. A., Hartmann, D. L., and Waugh, D. W.: Does the Holton-Tan Mechanism Explain How the Quasi-Biennial Oscillation Modulates the Arctic Polar Vortex?, J. Atmos. Sci., 69, 1713–1733, https://doi.org/10.1175/JAS-D-11-0209.1, 2012.

Naoe, H. and Shibata, K.: Equatorial quasi-biennial oscillation influence on northern winter extratropical circulation, J. Geophys. Res., 115, https://doi.org/10.1029/2009JD012952, 2010.

Takaya, K. and Nakamura, H.: A formulation of a phase-independent wave-activity flux for stationary and migratory quasigeostrophic eddies on a zonally varying basic flow, J. Atmos. Sci., 58, 608–627, 2001.

Zhang, J., Xie, F., Ma, Z., Zhang, C., Xu, M., Wang, T., and Zhang, R.: Seasonal Evolution of the Quasi-biennial Oscillation Impact on the Northern Hemisphere Polar Vortex in Winter, J. Geophys. Res., 124, 12 568–12 586, https://doi.org/10.1029/2019JD030966, 2019.

**Responses to Reviewer 2**

Wang et al investigate the influence of the QBO on total column ozone and stratospheric ozone. The authors confirm previous work on the role of the QBO for tropical and subtropical ozone. The main novelty of this paper is that it finds that the QBO at 20hPa has a zonally asymmetric imprint on subpolar ozone that is especially pronounced in DJF. This zonal structure occurs despite the QBO at 20hPa having a relatively weak impact on zonal mean stratospheric conditions. This result is not particularly surprising, but appears to not have been noticed before. A similar effect is also evident in a chemistry-climate model.

There are several major issues with the paper in its current form as described below. After these are addressed this paper should be publishable.

We thank the reviewer for the valuable comments and suggestions which helps to improve the manuscript substantially. We have revised the manuscript carefully based on the comments and suggestions of the reviewer and hope that the manuscript has been improved significantly. More details can be found in the revised manuscript as well as the point-to-point response as follows. The comments are shown in black and our replies are marked as blue.

Major comments:

1. I found the stippling on the plots that are intended to indicate statistical significance confusing. On most figures, regions with no discernable anomaly are still stippled, while the strongest anomalies are often not stippled at all. The simplest explanation is that there is a bug somewhere, however I apologize if I misunderstood something.

We have checked the code carefully, and there is not any bug in the code. In the region with strong anomalies, the variability is also large, which makes it hard to pass the statistical significance. For example, the standard deviation and the QBO signals of geopotential height (Z) are shown in the figure R4. The standard deviation of geopotential height is very large during DJF in the Arctic and during JJA and SON in the Antarctic, which makes the strong geopotential height anomalies not statistically significant.

[Figure]

**Figure R4.** Influences of QBO (QBOW-QBOE) on global geopotential height (Z at 10 hPa) based on ERA5 data for the period 1979-2020. The standard deviation of geopotential height in each season is also shown (contour lines). (a) MAM. (b) JJA. (c) SON. (d) DJF. Stippled areas indicate results that are statistically significant over the 95% level, using the two-tailed Student's t-test.

2. The key results of this paper appear to be only significant at the 90% level, if I understand the paper correctly. This is a fairly low bar. Would all significance in polar regions go away if the threshold was raised to 95%? Relatedly, it is surprising that the zonal structure in Figure 3d (in DJF when zonal structure is strongest) is not significant while it is in the annual average in Figure 2. Presumably this is because there is more variability in DJF, but this just begs the question as to how robust this zonal asymmetry truly is. In particular there is no clear explanation as to why this particular phase of the QBO should have the effect on Z* that it appears to have had over these ~40 years, and so I'm skeptical that additional data will necessarily support the authors conclusions. That being said, the model runs help demonstrate robustness.

We thank the reviewer for the valuable comments. We have updated all the figures to raise the significance to the 95% threshold, and the results do not change that much. We apologize for choosing the 90% level in the last version of the manuscript. As the reviewer indicated, the variability of TCO in DJF is strong, especially over the regions where the QBO related anomalies are strong. This can be seen from the standard deviation of TCO in different seasons as shown in Figure R5. The other possible reason is that Figure 2 used monthly anomalies with data samples of 492, while Figure 3 used seasonal mean with only 41 data samples, which will reduce the freedom of the significance test.

To further show the robustness of the results, we show the corresponding QBO signals of TCO in our Natural and NOQBO simulations in Figure R6. With a longer period of 145 years, the TCO anomalies associated with QBO in the Natural simulation are more significant. The robust impact of QBO on TCO can be further confirmed by the large difference between the Natural and NOQBO simulations. While the QBO is not nudged in the NOQBO simulation, the signals shown in the Natural run disappear.

[Figure]

**Figure R5.** Influences of QBO (QBOW-QBOE) on global total column ozone (TCO) in different seasons based on MSR2 data for the period 1979-2020. The standard deviation of TCO in each season is also shown (contour lines). (a) MAM. (b) JJA. (c) SON. (d) DJF. Stippled areas indicate results that are statistically significant over the 95% level, using the two-tailed Student's t-test.

[Figure]

**Figure R6.** Influences of QBO (QBOW-QBOE) on global total column ozone (TCO) in different seasons from the Natural (left) and NOQBO (right) simulations for the period 1955-2099. (a, e) MAM. (b, f) JJA. (c, g) SON. (d, h) DJF. Stippled areas indicate results that are statistically significant over the 95% level, using the two-tailed Student's t-test.

3. The dynamical explanation in Section 3.4 (lines 244-247) needs further refinement. Specifically, why exactly is a local ridge associated with more ozone, and a local trough with less ozone, in Figure 11? If it was just meridional advection, then the ozone anomalies should be collocated with the nodes of the height pattern, not the extrema.

We thank the reviewer for the good comments. The positive geopotential height anomalies in eastern North America and western Eurasia, and the negative anomalies over other regions of the Arctic, indicates a shift of the polar vortex. While the polar vortex acts as a barrier that damps the meridional transport and mixing between the polar region and the midlatitudes, it is very cold and the ozone concentrations are very low in the polar vortex. This shift of the polar vortex therefore leads to positive ozone anomalies

in eastern North America and western Eurasia and negative ozone anomalies in eastern Eurasia and the North Pacific. As the other reviewer indicated, temperature changes should be also considered for the ozone changes since the chemical reactions are temperature dependent. We then added some discussion about the influences of the temperature-dependent chemical effects. As shown in Figure R7, there are negative temperature anomalies collocated with the local trough. In the polar region, cold temperature anomalies may lead to more ozone destruction and subsequent negative ozone anomalies. Therefore, ozone anomalies may be caused by a combined effect of dynamical transport and temperature-dependent chemical reactions. We have added some discussions in the revised manuscript.

[Figure]

**Figure R7.** Influences of QBO (QBOW-QBOE) on the global temperature at 10 hPa in different seasons based on ERA5 data for the period 1979-2020. The climatology mean of temperature at 50 hPa in each season is also shown (contour lines). (a) MAM. (b) JJA. (c) SON. (d) DJF. Stippled areas indicate results that are statistically significant over the 95% level, using the two-tailed Student's t-test.

4.Much of the discussion and many of the figures more or less confirm earlier published work. (I'm specifically referring to the tropical and subtropical impacts of the QBO.) In this reviewer's opinion these figures can be moved to supplemental material, in order to focus more on the novel results.

We agree with the reviewer that there are some figures and discussions about the tropical and subtropical impacts of the QBO similar to earlier published work. We have reduced some of the discussions. However, including these figures would be helpful for the readers to understand the impacts of QBO from the tropics to extra-tropics and from zonal

mean to zonal structures. In addition, the other reviewer shows interests in and has some comments about the zonal mean features of the QBO impacts. We are sorry but hope to keep the figures in the main text.

Minor comments:

1.  There are two papers the authors appear to have not cited that are relevant to zonal asymmetries in the polar response to the QBO: Silverman et al 2018 and Elsbury et al 2021. While the focus in the current work differs from these paper, these papers should be discussed

    Thank you very much for the important information. The papers help us a lot to further understand the underlying mechanism. We have cited the two papers and added some discussion in the revised manuscript.

2.  Line 39-40: It is unclear what is the precise mechanism whereby the QBO affects the polar vortex. Garfinkel et al 2012 find evidence for a different mechanism though it is still unclear which mechanism is most important. This is discussed in the Elsbury et al paper

    Sorry for the inaccuracy description. We have updated the description as follows:

    "Such changes in zonal winds modify the vertical propagation of planetary waves and influence the strength of the polar vortex as well as the Brewer-Dobson circulation (BDC) according to the Holton-Tan mechanism (Holton and Tan, 1980, 1982; Watson and Gray, 2014; Zhang et al., 2019; Baldwin et al., 2019) or the QBO implicit meridional circulation mechanism (Garfinkel et al., 2012; Elsbury et al., 2020), and therefore play an important role in determining the dynamical circulation in the whole stratosphere (Naoe and Shibata, 2010; Garfinkel and Hartmann, 2011a, b; Anstey and Shepherd, 2014; Andrews et al., 2019; Zhang et al., 2020)."

3.  There are numerous technical edits that need to be made. Please send the paper to an English editor.

    We are sorry for that. However, it is not easy for us to find a native speaker to help us with the manuscript. We have checked the whole manuscript carefully from sentence to sentence and hope the text has been improved significantly.

4.  Line 43 compositions -> trace gases.

    Corrected.

5. Line 53: the details of where the peaks lay depends on the level used to define the QBO

   Thanks. We have modified the sentence as suggested.

6. Line 59 how are global patterns of ozone important for regional health? Please revise.

   Sorry for the inaccuracy description. We have updated the description as follows:

   "While the global pattern of ozone changes is important to the regional UV radiation as well as weather and climate, it is therefore interesting to look through the zonal differences of QBO in ozone."

7. Line 189-190 This discussion implies that the upper stratospheric ozone anomaly is dynamically driven and not photochemically driven. Please provide additional evidence/discussion as to whether photochemical processes are indeed not important

   Sorry for the misunderstanding here. Photochemical processes may also contribute to the ozone anomalies here. We have revised this sentence and added some discussions here.

8. Line 233-234 implies a specific direction of causality between T and vertical wind anomalies. While the statement is clearly true, the direction of causality is not necessarily clear, as both the T and w responses are fundamentally linked to the wind shear via thermal wind balance and mass continuity.

   Thanks for the good comments. We have revised this sentence as follows:

   "This is possibly related to the anomalously strong upwelling of the BDC in the tropics as seen in Fig. 6 and subsequent dynamical cooling."

Elsbury, D, Peings, Y, Magnusdottir, G. Variation in the Holton–Tan effect by longitude. Q J R Meteorol Soc. 2021; 1767– 1787. https://doi.org/10.1002/qj.3993

Silverman, Vered, Nili Harnik, Katja Matthes, Sandro W. Lubis, and Sebastian Wahl. "Radiative effects of ozone waves on the Northern Hemisphere polar vortex and its modulation by the QBO." Atmospheric Chemistry and Physics 18, no. 9 (2018): 6637-6659.

Garfinkel, C.I., Shaw, T.A., Hartmann, D.L. and Waugh, D.W., 2012. Does the Holton–Tan mechanism explain how the quasi-biennial oscillation modulates the Arctic polar vortex?. Journal of the Atmospheric Sciences, 69(5), pp.1713-1733.

We thank the reviewer for the important information. We have read through the papers carefully and cited these references in the revised manuscript.

---

## Author Response (AR2)

**Responses to Reviewer 1**

This is my second review of Wang et al. One of my initial comments was that I found the stippling on the plots that are intended to indicate statistical significance confusing. On most figures, regions with no discernable anomaly are still stippled, while the strongest anomalies are often not stippled at all. One possible explanation is that there is a bug somewhere, however the authors are confident there is no such bug. I am not convinced. For example, Figure 2 indicates that the tropical TCO anomalies are not significant, while the much weaker anomalies in midlatitudes are. Similarly, Figure 9 implies that the wind anomalies in the tropics are not significant, however this is a QBO composite so how is that even possible?

I must admit that I cannot recommend acceptance of this paper with the figures in their current state, as they fail to pass a basic sanity check. If the authors are still confident in their results, I suggest that the editor ask for an additional reviewer.

I did not re-read section 3.4 as once I saw figure 9 I stopped reading the paper, however my major comments on earlier sections were addressed reasonably, and the ACP copy editor should be able to fix the word usage issues.

We thank the reviewer for the very helpful comments. Yes, the reviewer is correct. We finally find the bug in the codes for testing the significance of the results. We are really sorry for the mistake. We have corrected all the figures and revised the corresponding descriptions in the whole manuscript. More details of the revision can be found in the revised manuscript as well as the point-to-point response as follows. The comments are shown in black and our replies are marked in blue.

Minor comments:
Line 82 implies the TCO data is from 1984-2020 while line 77 implies 1979-2020. Please clarify

We apologize for the confusing description. We used three types of data, including TCO data (with dimensions of longitude and latitude), zonal mean data with different vertical levels (with dimensions of latitude and altitude) and the 3-D data (with dimensions of longitude, latitude and altitude) in the analysis. The TCO data is available from 1979 to 2020, while the zonal mean data with different vertical levels is available from 1984 to 2020 and the 3-D data is only available for the period 2002-2020. We have added more details of the data in the revised manuscript to make it clear.

Line 133 Please clarify the units for the PC1>0.5 and PC1<-0.5 (I assume standard deviations is the unit, but this should be stated)

Yes, the unit is the standard deviation. We have added it in the revised manuscript.

Figure 4 are the units ppmv or ppm by mass?

The unit is ppmv. We have updated it in the revised manuscript.

**Responses to Reviewer 2**

General comments
I understand that the authors made many efforts to improve the manuscript. Thank you for considering my comments and suggestions. However, I have still found a lot of errors, inconsistencies, typos, and insufficient explaining of logic in the revised manuscript; thus, I have several questions and comments (see the minor comments). I do not think the current version of the manuscript is acceptable as it is; the manuscript still seems to need corrections and refinements.

We thank the reviewer for the further comments. We have revised the manuscript carefully based on the comments and suggestions of the reviewer and hope that the manuscript has been improved significantly. More details of the revision can be found in the revised manuscript as well as the point-to-point response as follows. The comments are shown in black and our replies are marked in blue.

Major comments
First, I apologize to the authors for making comments and questions on points I did not mention in the original version. I have checked the manuscript more carefully in this round of review, and I have further comments and questions.

We thank the reviewer for the further comments which are very helpful to improve the manuscript.

According to the description in the Method section (2.4), QBOW is a phase with westerly winds around 10 hPa. Is it correct? If so, for me, and possibly some others, who are used to the definition of QBO at 50 hPa, the relationship between the QBO phase and TCO anomaly sounds reversed and may be confusing. Would you add some descriptions explaining this? For example, "…during QBOW phases (easterly around 50 hPa), as compared with QBOE (westerly around 50 hPa).

The QBO index used in this study is the first principal component of the Empirical Orthogonal Function (EOF) analysis on the equatorial zonal wind in the stratosphere (70-10 hPa), which is synchronized with the 20 hPa equatorial zonal wind with a correlation coefficient of 0.99. We thank the reviewer for the good suggestion and have added some descriptions correspondingly in the revised manuscript.

The authors' analyses were performed by evaluating the difference between the composites of QBOW and QBOE (QBOW–QBOE), and they gave descriptions like "Zonally asymmetric features are seen in the polar regions…". I think they should state more clearly "zonally asymmetric features of the difference".

We thank the reviewer for the good suggestion. We have updated the descriptions to "zonally asymmetric features of the QBO signals (QBOW-QBOE)" or "zonally asymmetric QBO signals (QBOW-QBOE)" to make it more clearly.

Making the difference between QBOW and QBOE is a standard practice to examine QBO effects, but my concern is that in Figs. 9, 10, 11, 12, and 13b, the contours are

climatological means, while the colors are the difference between QBOW and QBOE. If the figures depicted the contours of QBOE means, I could easily understand how the QBOW changed the fields from QBOE; or if they used colors to show the difference between QBOW and the climatology along with the climatology contours, it would also be easy to understand the physical meaning. Do the authors assume that the climatology is near the mid-point between QBOW and QBOE?

Thanks for the good comment. We have changed the contours from climatological mean to QBOE means in the corresponding figures.

In association with this, I am confused with the results shown in Fig. 12, although this figure seems to be interesting. The authors state that the QBO related wavenumber-1 anomalies (QBOW–QBOE) are generally out of phase with the climatological pattern. Therefore, in the QBOW phase, the zonal asymmetry is smaller and it is expected that wave activity is less than that in the QBOE. Is this correct? My understanding of E-P flux anomaly associated with the QBO phases is that more wave flux goes into the high latitude stratosphere when the tropical zonal wind around 50 hPa is easterly, thus polar vortex is weaker, and vice versa (less wave flux goes into the high latitude stratosphere when the tropical zonal wind around 50 hPa is westerly, thus polar vortex is stronger.) This relationship is the Holton and Tan relationship and several other works also showed this. I am wondering if the anomalies in Fig. 12 is consistent with the results from the preceding studies based on the zonal-mean zonal wind and E-P flux. Would you represent Fig. 12 by making difference between QBOW and climatology (QBOW–climatology) and between QBOE and climatology (QBOE–climatology), and depict the differences from by colors? Then you can check whether the anomaly distribution is "in-phase" or "out of phase" compared to the climatological mean geopotential height distribution for each QBO phase. I think that "out of phase" reduces the zonal asymmetry and corresponds to less E-P flux, then zonal winds in the tropics and 50 hPa should be westerly (QBOE in the author's definition?).

We thank the reviewer for the comments. Figures R1 and R2 show the anomalies in the QBOW and QBOE phases separately as the reviewer suggested. As shown in Figure R1, the anomaly is out of phase with the climatology from eastern North America to western Europe and from eastern Eurasia to the North Pacific (about 60% of the focused area), but in phase with the climatology in other regions. Therefore, maybe it is inaccurate to say that the QBO-related wavenumber-1 anomalies (QBOW–climatology) are generally out of phase with the climatological pattern, since the phase shift of the anomaly compared to the climatology is more evident. For QBOE, the anomalies are opposite in sign with the QBOW (Figure R2) and therefore have more areas (about 60%) in phase with the climatology. We agree with the reviewer that there are less wave activities in the regions where the anomaly is out of phase with the climatology. However, this does not mean our results are inconsistent with the Holton-Tan mechanism. The QBO index used in this study (with a correlation of 0.99 to the U at 20 hPa) is not simply the opposite of the zonal winds at 50 hPa in the tropics (with a correlation of -0.18). Figure R3 shows the geopotential height anomalies during the QBOE using the 50 hPa U index, which is significantly different from the results using

the QBO index in this study during QBOW (Figure R1a). The geopotential height is anomalously high (indicating a weaker polar vortex) during QBOE (at 50 hPa) due to the Holton-Tan mechanism, with neglectable zonal asymmetry features. The anomalies with different wave numbers (Figures R2b-d) can not fully explain the overall anomalies (Figure R3a). Note that as introduced in the Method section, the sample size of QBOW and QBOE is nearly equal to each other in our QBO index, while the QBOW size is usually much larger than the QBOE size using the 50 hPa QBO index (Fig. 1). This may be one possible reason of the different zonal asymmetric features using different QBO index, however, the exact reason awaits further studies is out of the scope of this study.

[Figure]

**Figure R1. (a)** Influences of QBOW (QBOW-climatology) on geopotential height (Z at 10 hPa) in the northern hemisphere winter (DJF) based on ERA5 data for the period 1979-2020. **(b-d)** The corresponding changes of geopotential height associated with QBO in wave numbers 1-3. The climatological values of geopotential height in winter as well as the climatological patterns of wave numbers 1-3 are also shown (contourlines). Stippled areas indicate results that are statistically significant over the

[Figure]

**Figure R2. (a)** Influences of QBOE (QBOE-climatology) on geopotential height (Z at 10 hPa) in the northern hemisphere winter (DJF) based on ERA5 data for the period 1979-2020. **(b-d)** The corresponding changes of geopotential height associated with QBO in wave numbers 1-3. The climatological values of geopotential height in winter as well as the climatological patterns of wave numbers 1-3 are also shown (contourlines). Stippled areas indicate results that are statistically significant over the 95% level, using the two-tailed Student's t-test.

[Figure]

**Figure R3. (a)** Influences of QBOE (QBOE-climatology, using the 50 hPa U as the QBO index) on geopotential height (Z at 10 hPa) in the northern hemisphere winter (DJF) based on ERA5 data for the period 1979-2020. **(b-d)** The corresponding changes of geopotential height associated with QBO in wave numbers 1-3. The climatological values of geopotential height in winter as well as the climatological patterns of wave numbers 1-3 are also shown (contour lines). Stippled areas indicate results that are statistically significant over the 95% level, using the two-tailed Student's t-test.

An interesting point is the phase shift of the anomaly compared to the climatology in Fig. 12. This seems to be a new finding. I hope the authors represent this for the anomalies in the QBOW and QBOE phases separately with the climatology contours and discuss the reasons.

As shown in Figures R1-R2, the phase shift of the anomaly compared to the climatology is evident for both QBOW and QBOE phases. The reason for the phase shift may be explained by the wave activity changes as shown in Figure R4 (Figure 13 in the main text). Note that there are some differences between the current version of Figure 13 and the last version. This is because we used the 2-D (horizontal) divergence of the wave

fluxes in the former version, but use the 3-D (both horizontal and vertical) divergence of the wave fluxes in the current version, which is more reasonable since the vertical propagation of waves and its divergence is very important. Compared to QBOE phases, more waves propagate upward from the troposphere to the stratosphere over the eastern Eurasia and North Pacific sector (60°E to 120°W, red contour lines in Fig. 13a) of the Arctic (north of 70°N), but less waves propagate upward in other sectors of the Arctic (blue contour lines in Fig. R4a) during QBOW phases. The favorable upward propagation of planetary waves over eastern Eurasia and the North Pacific may be due to the relatively large climatological wave flux from the troposphere to the stratosphere in these regions (Elsbury et al., 2021). This leads to a weakening of the zonal wind in the eastern Eurasia and North Pacific sector but an enhancement of zonal wind in other sectors of the Arctic due to the wave-mean flow interactions. To conserve angular momentum and maintain mass continuity (Kidston et al., 2015), the weakening (strengthening) of the zonal wind near the pole leads to stronger (weaker) westerlies in the subpolar regions (50-70°N) over eastern Eurasia and the North Pacific (other sectors). On the other hand, 3-D waves diverge in the eastern Eurasia and North Pacific sector but converge in other sectors (shading in Figure R4a), which also contribute to the stronger (weaker) westerlies in the subpolar regions (50-70°N) over eastern Eurasia and the North Pacific (other sectors) due to the wave-mean flow interactions. This indicates a shift of the polar vortex in the subpolar regions from North America and the North Atlantic to eastern Eurasia and the North Pacific, which is consistent with the geopotential anomalies as shown in Figure 12 in the main text and also Figure R1. This shift of the polar vortex is also consistent with previous studies (e.g., Elsbury et al., 2021). We hope this can help to explain the phase shift of the anomaly compared to the climatology. However, why the changes of planetary waves associated with QBO show varying characteristics over different regions awaits further studies.

[Figure]

**Figure R4.** Influences of QBO (QBOW-QBOE) on T-N wave flux **(a)** and zonal winds

**(b)** at 10 hPa north of 30° N during winter (DJF) based on ERA5 data for the period 1979-2020. In **(a)**, the meridional and zonal components of the wave flux are shown as vectors, the vertical component is shown as contour lines (positive in red) and the divergence of the wave flux is shaded. In **(b)**, the climatological values of zonal wind during QBOE are shown in contour lines (solid lines for westerly) and the anomalies (QBOW-QBOE) are shaded. Stippled areas indicate results that are statistically significant over the 95% level, using the two-tailed Student's t-test.

Another issue is that there are several sentences in which it is difficult to understand the logic or physical meaning. For example, the authors should carefully explain that the data period of TCO (1979–2020) and that of ozone at 10 hPa (2002–2020) is different when they compare between Figs.8 and 11.

We apologize for the confusing information. The different period of analysis is due to the data availability. The TCO data is available since 1979, while the satellite observed ozone with global distribution at different levels is only available since 2002. We have added more information in the Data and Method Section to make it clearer.

Finally, I found an inconsistency in the figure format between Figs. 2, 3, 7, and 8 and Figs. 9–11

Sorry for the inconsistency. We have changed the format of the Figs. 9-11 to the same format as Figs. 2, 3, 7, and 8.

Minor comments
The line numbers refer to the ATC (article tracked changes) version.

1. Abstract: It would be better to include the period for analysis (1979–2020) somewhere. The ERA5 data have recently been updated for the period before 1979 due to some problems affecting the performance in the tropics in the original version. However, this would not affect the case under consideration, since the target period of this article is 1979–2020.

As described in the Data and Methods Section, the periods for analysis are different for TCO (1979-2020), zonal mean (1985-2020) and 3-dimensional ozone due to the data availability. Therefore, we are sorry but cannot simply add a period for analysis here.

2. Abstract: An explanation of the QBO definition is needed, otherwise readers may be confused. See minor comment No.10.

Thanks for the good suggestion. We have added the QBO definition in the Abstract in the revised manuscript.

3. Lines 17–18, "influenced by the corresponding temperature changes and subsequent chemical reactions": It seems better to say, "influenced by chemical reactions associated with the corresponding temperature changes."

Thanks. We have updated this sentence.

4. Lines 29–30: Since the description associated with the references is a general description, add "e.g.," before "Son et al., 2008….", or add more appropriate references (review papers, for example).

Thanks. We have added "e.g.," before "Son et al., 2008…." as the reviewer suggested.

5. Line 30, "WMO et al., 2018": Should be "WMO 2018."

Corrected.

6. Line 58: So, what is the level used to define the QBO?

We use a QBO index around 20 hPa in this study. Such information has been described in the Data and Methods Section and also added in the Abstract and Introduction in the revised manuscript.

7. Lines 64–65: I do not understand why the global pattern of ozone changes is important to the regional UV radiation. It is certain that regional ozone change is a cause of regional UV change.

Sorry for the confusing description. It should be "While the global pattern of ozone changes is important to regional variations of the UV radiation".

8. Lines 65–66, "it is therefore interesting to look through the zonal differences of QBO signals in ozone": I do not understand how this flows logically from the previous sentence.

We try to highlight the importance of the zonal differences of QBO signals in ozone, however, it seems not successful. Therefore, we decide to remove the whole sentence in the revised manuscript.

9. Line 144, "deviation of the zonal mean": Change to "deviation from the zonal mean"

Corrected.

10. Lines 148–149, "monthly anomalies of TCO near the equator (10S-10N) are anomalously high during QBOW phases compared with QBOE.": This expression is understandable because the authors define QBO by PC1 (around 20 hPa). However, for me, and possibly some others, who are used to the definition of QBO at 50 hPa, the relationship between the QBO phase and TCO anomaly sounds reversed and may be confusing. Would you add some descriptions explaining this? For example, "…during QBOW phases (easterly around 50 hPa), as compared with QBOE (westerly around 50 hPa). I hope this kind of explanation may be repeated in other places where "QBOW" and "QBOE" appear in the text.

Thanks for the good suggestion. We have added such information in the corresponding places in the whole manuscript as suggested.

11. Line 156: Would you state here or somewhere that the anomaly distribution is the difference between QBOW and QBOE? The authors often mention a "zonally asymmetric feature" or "positive/negative anomaly is seen in …" in the text, but it is

not compared to a climatological mean field but to the field in the QBOE.

Thanks. We have defined the QBO-related anomalies/signals clearly as the differences between QBOW and QBOE (QBOW-QBOE) in the Data and Methods Section as well in the title and figure caption in every figure. We have added such information here in the revised manuscript as the reviewer suggested.

12. Line 157, "to our understanding": Change to "to the best of our knowledge."

Updated.

13. Line 164: Add "(Fig. 2d)" after "the QBO signals disappear in the NOQBO run"

Added.

14. Lines 164–165: I do not understand this sentence. My understanding is "Because the only difference between the two model simulations is the QBO nudging and because the difference in the two composites is similar between the simulation and the observation, this result indicates that the differences in the two composites of the observed TCO are mostly due to QBO."

Thanks for the suggestion. Yes, the description as the reviewer suggested is clearer. We have updated this sentence.

15. Line 181, "disappear": The phrase "not evident" would be better.

Updated.

16. Line 195, "in the stratosphere Fig. 4d": Do you mean "in the stratosphere in Fig. 4d"?

Yes, we have corrected this sentence.

17. Line 196, "sown": Typo.

Corrected.

18. Lines 196–198 and Fig. 5: Can you say this without statistical significance in the positive anomalies in the lower and middle stratosphere? The authors could use the 90% level figure in the previous version and explain the relatively low significance for the seasonal panels.

We are really sorry but there was a bug in our codes of the statistical significance test. Fig. 5 has been updated and the QBO-related signals are more significant at the 95% level.

19. Line 208, "SOLOMON et al.": Should be "Solomon et al."

Corrected.

20. Line 217, "to our understanding": Change to "to the best of our knowledge."

Updated.

21. Lines 219–221: Fig. 7 indicates the distribution of QBOW–QBOE. The asymmetry during QBOE is not mentioned.

Thanks. We have updated this sentence.

22. Lines 259–261: I think the authors should say, "The results are consistent with preceding studies, considering the different definition of QBO."

We thank the reviewer for the good suggestion. We have updated this sentence in the revised manuscript.

23. Lines 271–272: 10 hPa is a difficult altitude level to use to discuss the dominance of the chemical process or dynamical (transport) process for ozone, because it is a transition altitude from dynamical control at lower altitudes (except for polar lower stratosphere) to chemical control to upper altitudes. For example, in panels (d) in Figs. 9 and 10 (DJF), the ozone and temperature anomalies have opposite signs in the SH mid and high latitudes, and this is considered to result from chemical control in the austral summer. In some regions of the NH high latitudes in DJF, the anomalies have the same sign, and this is considered to result from dynamical control in boreal winter (the negative anomalies indicate less heat and ozone transport from the midlatitudes).

We thank the reviewer for the very useful information. We have added the discussions mentioned above in the discussion about the relationship between temperature and ozone anomalies.

24. Line 275, "anomalies at 10 hPa during QBOW phases": These are anomalies from QBOE, that is, QBOW-QBOE.

Updated.

25. Lines 277–279: Because Fig. 11 shows the Z difference between QBOW and QBOE, zonal asymmetry of Z in the QBOE should be mentioned.

Thanks, we have updated the descriptions here.

26. Lines 287–288, "However, ozone anomalies are all positive over the Antarctic from the merged satellite data": Which figure are you referring to? Fig. 8b? I think the data period is different between Fig. 8 and Fig. 11.

Yes, we are referring to Fig. 8b here and the data period is indeed different between Fig. 8 and Fig. 11 due to the data availability. We have added some discussions here in the revised manuscript.

27. Lines 287–290: The authors should consider and discuss the difference in data periods (2002–2020 for Fig. 8 and 1979–2020 for Figs. 11 and S6).

Thanks, we have added some discussions about this issue in the revised manuscript.

28. Line 292, "during QBOW": The distribution is an anomaly from QBOE.

Updated.

29. Lines 303–304 and 305–306: The anomalies indicated by color are those from QBOE, not from the climatology.
Updated.

30. Lines 321–328: I understand the necessity of showing the 3-D QBO anomaly in the Northern Hemisphere in DJF in Fig. 12, because in winter, planetary wave activity is active. Why did the authors explain the QBO anomalies in the NH and SH in SON? Is there nothing worth to mention in MAM? There is a description in lines 347–348 that seems to be the answer. Please mention it in Section 3.4.
We thank the reviewer for the good suggestion and have added such information as suggested in the revised manuscript.

31. Lines 332–335: Please mention the QBO phase.
Updated.

32. Line 336, "a single-factor controlling simulation": It would be better to say, "a sensitivity simulation."
Updated.

33. Line 340, 120W-30E: I found this longitude range in Conclusions and in the Abstract (line 12) but could not find it in the other sections. Is it inconsistent?
We have also added the longitude range in the main text while describing the corresponding figures.

34. Lines 469–470: Should be "WMO (World Meteorological Organization), Scientific Assessment of Ozone Depletion: 2018, Global Ozone Research and Monitoring Project–Report No. 58, 588 pp., Geneva, Switzerland, 2018."
Corrected.

35. Figs. 9–11: Why is the format for these figures different form that for Figs. 2, 3, 7, and 8?
There is not any special reason for that. We have changed the format of Figs. 9-11 to keep consistent with Figs. 2, 3, 7, and 8.

References:
Elsbury, D., Peings, Y., and Magnusdottir, G.: Variation in the Holton-Tan effect by longitude, Quart. J. Roy. Meteor. Soc., 147, 1767–1787, https://doi.org/10.1002/qj.3993, 2021.
Kidston, J., Scaife, A. A., Hardiman, S. C., Mitchell, D. M., Butchart, N., Baldwin, M. P., and Gray, L. J.: Stratospheric influence on tropospheric jet streams, storm tracks and surface weather, Nat. Geosci., 8, 433–440, https://doi.org/10.1038/NGEO2424, 2015.

---

## Author Response (AR3)

**Comments to the author**:

In your response and revision you seem to have resolved the primary concern of Reviewer 1 regarding the depiction on the Figures of regions where signals are statistically significant. My impression is also that you have addressed most of the comments of Reviewer 2. Therefore I do not see it as appropriate to request further comments from the Reviewers regarding your revision. However there are a number of minor points, mostly arising out of comments from Reviewer 2, that I request that you address before publication. (Please provide with your revised paper a brief set of responses that make it clear which of these points you have addressed and which you have not, with brief justification for the latter.)

We thank the editor very much for the further comments and suggestions. We have addressed the points of 2, 4, 5 and 6 as the editor suggested. For points 1 and 3 regarding to the QBO definition, we have considered the problem carefully and we would like to solve the problem as follows: 1) describe the definition of the QBO index used in this study clearly in Section 2.4 and explain the reason of the choice; 2) remove the construction like 'QBOE (westerly at 50hPa); 3) discuss the results of using the PC2 (in effect the wind direction at 50 hPa) as the QBO index in Section 4. Then we think that the audience will not be confused with the results shown in this study and can also compare the results to previous studies.

More details of the revision can be found in the revised manuscript as well as the point-to-point response as follows. The comments are shown in black and our replies are marked in blue.

1) A legitimate concern of Reviewer 2 was that your definition of QBOW vs QBOE (in effect the wind direction at 20hPa) would cause confusion since the vast majority of papers on this topic define QBOW vs QBOE by the wind direction at 50hPa. You have addressed this, as suggested by the referee but writing 'QBOE (westerly around 50hPa)' etc. But have you considered the simpler approach of simply swapping your definition of QBOW and QBOE? You can explain that the phase is defined by the EOF and that your definition is equivalent to using the sign of the wind direction at 50Pa (or some other nearby pressure level if you want to be more exact).

We thank the editor for the good suggestion. However, as the editor noticed in point 3, the QBO index used in this study is not simply the opposite of the zonal winds at 50 hPa/70 hPa in the tropics (with a correlation of -0.18/-0.62). Therefore, a swapping of the QBOW and QBOE is not equivalent to using the sign of the wind direction at 50 hPa/70 hPa. We will describe our solution to this problem in the response to point 3.

2) Contours: you are in Figs 9-12 now using contours to show QBOE means. That is fine. But please can you note the contour interval used explicitly in the each of the relevant Figure captions. (The interval can be deduced, but it would be helpful for the reader if it was simply given.)

We thank the reviewer for the good comment. We have added the contour interval in the figure captions in Figures 9-12 as suggested.

3) Further issue regarding QBO phase: As I note above, you have used the construction 'QBOE (westerly at 50hPa)' -- and I have suggested an alternative to that -- but then in your long response to Reviewer 2 regarding Figure 12 you say 'The QBO index used in this study (with a correlation of 0.99 to the U at 20 hPa) is not simply the opposite of the zonal winds at 50hPa in the tropics (with a correlation of -0.18)'. So this suggests that the 'QBOE (westerly at 50hPa)' construction is simply misleading, in which case you should not be using it and you need to find some other way of expressing the relation of your definition of QBO phase to others that have commonly been used, with the wind at 50hPa being the most common. I might ask why you chose this particular definition of the QBO -- I don't believe that you explain that. Certainly the use of different definitions of the QBO in different papers is potentially confusing and what is important is that the choice of the definition is justified (perhaps the signals are weaker when other definitions are used) and there is some effort to explain who the choice affects the relation of the results to those presented in other papers on this topic. At present you seem to have decided on a way (suggested by Reviewer 2) to state the relation of your own definition of QBO phase to the definition that is most often used by others (wind direction at 50hPa) but then you are saying that this statement is not a valid one.

We agree with the editor that the construction like 'QBOE (westerly at 50hPa) is misleading. As described in Section 2.4, our choice of the QBO index is due to 2 reasons: 1) PC1 (in effect the wind direction at 20 hPa) is close to the middle stratosphere (~10 hPa), where the ozone mixing ratios are highest; 2) The sample size of QBOW and QBOE is nearly equal to each other, while the QBOW size is usually much larger than the QBOE size using PC2 (Fig. 1). After careful consideration, we would like to solve the problem as follows: 1) describe the definition of the QBO index used in this study clearly in Section 2.4 and explain the reason of the choice; 2) remove the construction like 'QBOE (westerly at 50hPa); 3) discuss the results of using the PC2 (in effect the wind direction at 50 hPa) as the QBO index in Section 4. Then we think that the audience will not be confused with the results shown in this study and can also compare the results to previous studies.

Figure R1 shows the influences of QBO (QBOW-QBOE, using PC2 as the QBO index to indicate U at 50 hPa) on global total column ozone (TCO) in different seasons based on MSR2 data 1979-2020. In general, there are also some zonally asymmetric features in the differences of TCO between QBOW and QBOE phases and the magnitude of the anomalies are comparable to that shown in Figure 3 in the main text. In DJF, the QBO signals using PC2 are opposite in sign with that using PC1 in the Northern Hemisphere (NH), while the zonal asymmetry is not that obvious. In MAM, the TCO anomalies are all negative in most of areas in the mid-to-high latitudes of the NH no matter which QBO index is used. In JJA, the PC2 related TCO anomalies are in the same sign with the PC1 related anomalies in the NH, but opposite in sign with the PC1 related anomalies in the Tropics and the Southern Hemisphere (SH). In SON, the zonal asymmetry of the PC2 related TCO anomalies is more obvious in the SH but less significant in the NH compared with PC1. It is very interesting that there are significant differences in the QBO related signals while using QBO index at different levels. We would like to leave these questions, e.g., why the PC2 related TCO anomalies are more zonal asymmetry during SON in the SH but less zonal asymmetry

during DJF in the NH, as open questions and try to find an explanation in future studies. We have added Figure R1 in the Supplement and discussed about the results in Section 4 in the revised manuscript.

[Figure]

Figure R1. Influences of QBO (QBOW-QBOE, using PC2 as the QBO index to indicate U at 50 hPa) on global total column ozone (TCO) in different seasons based on MSR2 data 1979-2020. (a) MAM. (b) JJA. (c) SON. (d) DJF. Stippled areas indicate results that are statistically significant over the 95% level, using the two-tailed Student's t-test.

4) Period of analysis: Reviewer 2 asked that the period of the analysis be given in the abstract -- your response is that is not possible because there are three different periods of analysis corresponding to three different data types. But surely it is perfectly possible to state that in one sentence in the abstract.
Thanks. We have added a sentence for the period of analysis in the abstract.

5) There is an error in equation (1) -- there should not be a 'z' subscript on the capital Theta '.
We are really sorry for the mistake and thank the editor for the correction. We have corrected the equation in the revised manuscript.

6) Please give the length of the model integrations (I think 1979-2020) in the text in Section 2.4.
Sorry for the missing information. The model simulations were integrated from 1955 to 2099. We have added this information to the revised manuscript.